# Towards Self-Evolving Agent Benchmarks: Validatable Agent Trajectory via Test-Time Exploration

**Dadi Guo**[1,2*], **Tianyi Zhou**[1,3*], **Dongrui Liu**[1*], **Chen Qian**[4], **Qihan Ren**[5], **Shuai Shao**[5], **Zhiyuan Fan**[2], **Yi R. Fung**[2], **Kun Wang**[6], **Linfeng Zhang**[5], **Jing Shao**[1†]

[1]Shanghai Artificial Intelligence Laboratory  [2]Hong Kong University of Science and Technology
[3]University of Michigan    [4]Renmin University of China    [5]Shanghai Jiao Tong University
[6]Nanyang Technological University
`dguoae@connect.ust.hk`  `zhtianyi@umich.edu`
`{liudongrui,shaojing}@pjlab.org.cn`

## Abstract

Recent advances in large language models (LLMs) and agent system designs have empowered agents with unprecedented levels of capability. However, existing agent benchmarks are showing a trend of rapid ceiling-hitting by newly developed agents, making it increasingly difficult to meet the demands of evaluating agent abilities. To address this problem, we propose the **T**rajectory-based Validated-by-**R**eproducing **A**gent-benchmark **C**omplexity **E**volution (**TRACE**) framework. This framework takes an original task from an existing benchmark and encourages agents to freely explore and evolve it into a new task with higher difficulty while recording the corresponding execution trajectories. The framework proceeds in three stages: (1) *evolutionary proposal mining*, which generates task evolution proposals through preliminary exploration and divergent thinking; (2) *problem construction via free exploration*, where proposals are instantiated into concrete problem instances through agent exploration, with execution trajectories recorded along the process; and (3) *multi-level validation*, which ensures that the evolved tasks are accompanied by reproducible and logically coherent trajectories. Experiments on the GAIA benchmark demonstrate that the **TRACE** framework consistently enhances task complexity while improving correctness reliability through trajectory-level validation. In addition, our framework can successfully adapt to and improve reasoning benchmarks such as AIME-2024. This work marks a paradigm shift from static, manually curated benchmarks to dynamic, self-evolving evaluation systems, providing a sustainable and challenging foundation for agent development. Code and data can be found at `https://github.com/titanwings/trace-benchmark-evolving`.

## 1 Introduction

The paradigm of artificial intelligence is rapidly shifting towards autonomous agents capable of complex reasoning (Comanici et al., 2025; Huang & Yang, 2025), planning (Huang et al., 2024), and tool utilization (Qu et al., 2025; Wang et al., 2024a). This progress is starkly evident in the performance on challenging agent benchmarks (Mialon et al., 2023; Jimenez et al., 2023), which were once considered formidable. For instance, on the GAIA benchmark which is designed to test real-world assistant capabilities, top-performing agents have achieved scores exceeding 90% (GAIA Benchmark Team, 2025), rapidly closing the gap with the human baseline. This rapid pace signals an urgent challenge: existing benchmarks are becoming saturated, diminishing their ability to differ-

---

*Equal contribution. Work done during an internship at Shanghai Artificial Intelligence Laboratory, supervised by Dongrui Liu

†Corresponding author

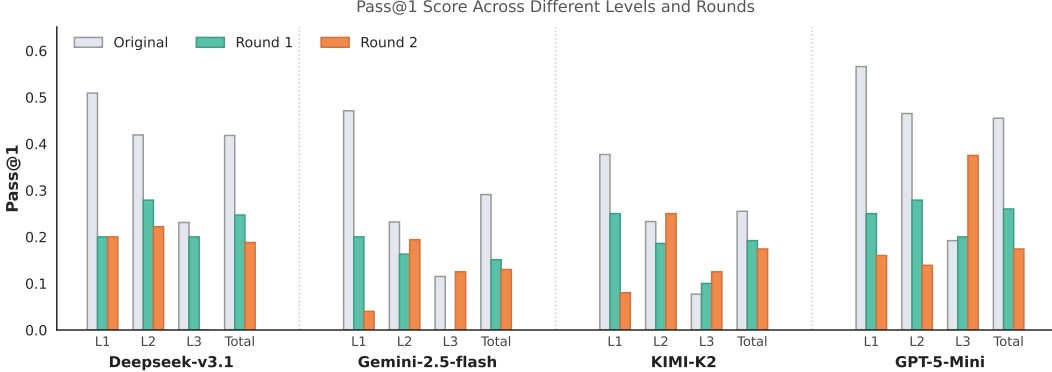

Figure 1: Model performance comparison on the Pass@1 metric across four distinct difficulty levels and evolution rounds under the **TRACE** framework. As the number of evolution rounds increases, the performance of models shows a downward trend, demonstrating that our framework successfully evolves more challenging tasks.

entiate state-of-the-art agents and risking progress being driven by overfitting to static test sets rather than generalizable intelligence. However, the cost of manually creating novel, complex, and reliable tasks is a labor-intensive, time-consuming, and expensive process, which highlights an urgent need for an automated and scalable approach to agent benchmark evolution.

However, evolving agent tasks presents unique challenges not found in conventional domains like mathematical reasoning (Hendrycks et al., 2021; Guo et al., 2025) or knowledge-based question answering (Yang et al., 2018). Agent tasks are defined by two key characteristics: (1) their procedural nature, which emphasizes complex, multi-step interactions with dynamic, scalable real-world environments (e.g., websites, APIs) (Huang et al., 2025; Liu et al., 2025a; Gao et al., 2025a), and (2) their immense diversity, spanning from web navigation to software operation. These characteristics render traditional task evolution methods, such as rule-based parameter mutation or scaling largely ineffective. For instance, rule-based changes (Wu & Liu, 2025; Wang et al., 2024b) (e.g., altering a specific keyword in a search query) are often insufficiently robust. In a dynamic web environment, such a minor change could break the task's solvability entirely rather than increasing its reasoning complexity. Similarly, merely scaling up a task (Liu et al., 2025b) (e.g., asking to book three flights instead of one) often increases repetition, rather than the core cognitive or planning challenges. Thus, a new paradigm is required that moves beyond superficial modifications to fundamentally enhance the procedural, logical and semantic complexity of agent tasks.

To bridge this gap, we introduce **TRACE** (**T**rajectory-based Validated-by-**R**eproducing **A**gent-benchmark **C**omplexity **E**volution). Departing from rigid, rule-based heuristics, TRACE leverages the agentic capabilities of LLMs to drive task evolution through three specialized components. The *Evolutionary Proposer* identifies diverse evolution directions via preliminary exploration; the *Exploration Executor* navigates real-world environments to materialize these proposals into recorded execution trajectories; and the *Trajectory Validator* enforces logical coherence and reproducibility. This ensures that evolved tasks are grounded in verifiable solution paths rather than relying solely on final-answer correctness.

The coordination between these agents enables a structured workflow for task evolution. This process begins with the *Evolutionary Proposer*, which analyzes task bottlenecks and explores its semantic space to identify evolutionary forks. Consider the question: *"What is the name of Taylor Swift's debut album?"* Rather than stopping at the direct answer, it examines related artifacts, such as the music video for the single *Teardrops on My Guitar*, and generates multiple evolution proposals. These proposals introduce additional reasoning dependencies, such as linking the video's actor to their filmography or tracing their cross-domain career trajectory. Subsequently, the *Exploration Executor* materializes these proposals through **proposal-guided trajectory construction**. It traverses the original solution path and integrates compatible proposals at appropriate intermediate states. For example, it may first inject a proposal to identify the actor in the video (*Tyler Hilton*) and the TV series he starred in (*One Tree Hill*), then introduce a second proposal that pivots to his music career. This process transforms a single-hop lookup into a multi-step reasoning task with cross-domain dependencies, culminating in a more complex final question (Trinh et al., 2024; Fang et al.,

2025; Sun et al., 2024; Gao et al., 2025b), such as: *"The male lead in the music video for a song from Taylor Swift's debut album is a multi-talented individual who also starred in a long-running TV series that premiered in 2003. What is the title of this individual's own debut studio album?"* More broadly, this bottleneck-aware proposal and injection process extends beyond information-seeking tasks to reasoning-intensive, mathematical, and coding problems, demonstrating the generality of the workflow. Importantly, each exploration step is recorded as an execution trajectory that serves as the basis for validation, ensuring that the introduced complexity remains transparent and verifiable.

This architecture balances flexibility and rigor in task evolution. Flexibility arises from the separation of the *Evolutionary Proposer* and the *Exploration Executor*, enabling evolution along multiple dimensions, such as deeper tool integration, increased logical dependency, or combinatorial composition, without reliance on hard-coded templates. To maintain rigor, the Executor outputs the evolved task together with its execution trajectory, including reasoning steps, tool invocations, and observed results. The *Trajectory Validator* then audits logical coherence and re-executes each step to confirm reproducibility. This validation loop ensures that generated benchmarks are complex, solvable, and grounded in executable evidence.

In essence, TRACE models the workflow of a human benchmark designer while replacing manual curation with an autonomous evolution process. Unlike static rule-based approaches, it enables agents to explore real-world environments (*e.g.*, the live internet) as structured task substrates. This exploration is guided by bottleneck analysis: by diagnosing limitations of a seed task such as shallow tool usage or limited reasoning depth, the framework steers evolution toward targeted structural augmentation. As a result, TRACE evolves tasks into tool-integrated, multi-step challenges that better reflect real-world problem-solving dynamics, repositioning the agent from a passive solver to an active benchmark constructor.

Our contributions are threefold:

- We propose **TRACE** (**T**rajectory-based Validated-by-**R**eproducing **A**gent-benchmark **C**omplexity **E**volution), a benchmark self-evolving framework that encourages agent exploration and records execution trajectories as first-class artifacts, ensuring transparency and reproducibility in task evolution.

- We empirically validate TRACE on general-purpose agent benchmarks such as GAIA as well as reasoning benchmarks such as AIME-2024, demonstrating that it consistently produces tasks of higher difficulty, on which prominent agent systems exhibit significant performance degradation shown in Figure 1, thereby validating the effectiveness of our approach.

- Beyond "one-more-hop" edits, our experiments reveal a *From Seed to Spark* pattern: under our TRACE framework, the model autonomously explores and evolves problems that can shift into entirely different capability domains (*e.g.*, from retrieval to math and coding), thereby substantially increasing the diversity of evolved tasks. The evolved tasks exhibit greater *task diversity* and require deeper *reasoning depth*, providing a robust methodology for rigorously evaluating and advancing future agents.

## 2 RELATED WORK

**Agent Benchmark** A range of agent benchmarks have been proposed to evaluate LLM-based agents under realistic problem-solving settings. GAIA (Mialon et al., 2023) focuses on human-centric questions that require reasoning, multimodal understanding, web browsing, and tool use. USACO (Shi et al., 2024) adapts competitive programming problems with unit tests and reference solutions to assess algorithmic reasoning. MLE-bench (Chan et al., 2024) evaluates end-to-end ML engineering by grounding tasks in Kaggle-style competitions spanning data preparation, model training, and experiment management. SWE-bench (Jimenez et al., 2023) measures software engineering ability through real GitHub issues that require generating patches under authentic repository contexts.

Beyond answer-verified benchmarks, several works emphasize *process-level* evaluation by analyzing agent trajectories. WebArena (Zhou et al., 2023) provides self-hosted websites with executable scoring and trajectory replay for reproducible step- and task-level analyses. Mind2Web (Deng et al., 2023) contributes human demonstrations on real websites with fine-grained metrics (e.g., element

accuracy and step success), and its multimodal extension (Pahuja et al., 2025) augments trajectories with multimodal signals and step-wise evaluation.

Despite their breadth, these benchmarks are largely *static*: tasks and environments are pre-defined and updated infrequently. As agent capability improves, static benchmarks can saturate and become costly to refresh at a cadence aligned with model iteration, motivating scalable approaches to benchmark evolution.

**Benchmark Evolving** Benchmark evolution has been explored to mitigate saturation of static datasets. Benchmark Self-Evolving (Wang et al., 2024b) and AutoEvoEval (Wu & Liu, 2025) apply predefined atomic operations to reformulate reasoning and close-ended QA tasks via structural or semantic perturbations, primarily targeting robustness rather than increasing procedural task complexity. AdamMeme (Chen et al., 2025b) adopts multi-agent refinement to iteratively update meme datasets for probing harmfulness-related reasoning, but is specialized to this domain. EvoCodeBench (Li et al., 2024) updates code benchmarks by periodically ingesting new repositories to reduce leakage, relying on curated data streams and fixed schedules rather than model-driven exploration.

In contrast, TRACE evolves agent tasks via *test-time exploration* in real-world environments. Instead of predefined edits or scheduled refreshes, agents autonomously construct harder tasks and output their execution trajectories, which are subsequently validated for logical coherence and reproducibility. This trajectory-aware evolution supports scalable benchmark updating while preserving solvability and verifiability.

## 3 PRELIMINARY

### 3.1 AGENTIC WORKFLOW AS A DAG

In this section, we represent an agentic workflow $W$ as a directed acyclic graph (DAG) $G = (\mathcal{S}, \mathcal{E})$, where the node set $\mathcal{S} = \{S_1, S_2, \ldots, S_N\}$ corresponds to an ordered sequence of discrete LLM-invoking steps, and the edge set $\mathcal{E} \subseteq \mathcal{S} \times \mathcal{S}$ encodes data dependencies and control-flow constraints. Specifically, an edge $(S_i, S_j) \in \mathcal{E}$ signifies that the output of step $S_i$ is required as input for step $S_j$. Each node can be abstracted as the quadruple below:

$$S_i = \big(c_{i-1}, \ r_i, \ a_i, \ o_i\big),$$

where

- **Context** $c_{i-1}$: the interaction history up to step $i-1$ (previous actions and observations together with fixed task/context).
- **Reasoning in test-time** $r_i$: the agent's latent reasoning state at step $i$ (internal scratchpad/planning variables and intermediate choices that are not executed by the environment).
- **Action** $a_i$: the external action/message emitted at step $i$ conditioned on $c_{i-1}$ and $r_i$ (e.g., a tool/API call with arguments, code to run, a retrieval query, or a user-visible response), which is the only component actually executed outside the agent. This state-transit process could be modeled as $p(a_i \mid c_{i-1}, r_i) = \pi_a\big(a_i \mid c_{i-1}, r_i\big)$.
- **Observation** $o_i$: the feedback returned by the environment after executing $a_i$ (e.g., tool outputs, retrieved documents, execution logs, state deltas, optionally a numeric score).

After execution, the context is updated as $c_i = c_{i-1} \oplus (a_i, o_i)$, and edges $(S_i, S_j)$ arise when artifacts from $S_i$, typically $o_i$ or the updated context $c_i$, are consumed by $S_j$; tool-free steps are a special case where $a_i$ is a user-facing message and $o_i$ may be empty.

### 3.2 EXPLORATION TRAJECTORIES FOR BENCHMARK EVOLUTION

Given the DAG formulation of an agentic workflow, we next define how *exploration trajectories* serve as the foundation for benchmark evolution. An *execution trajectory* $\tau$ is a path through the workflow DAG,

$$\tau = \big\langle S_1, S_2, \ldots, S_T \big\rangle,$$

where each $S_i$ is realized by the tuple $(c_{i-1}, r_i, a_i, o_i)$. Unlike static benchmarks that only verify the final output $o_T$, we treat the entire trajectory $\tau$ as a first-class artifact: it captures the agent's reasoning, tool-use decisions, and environment feedback across all intermediate steps.

Formally, we denote the trajectory distribution under an agent policy $\pi$ as

$$p_\pi(\tau) = \prod_{i=1}^{T} \pi_a(a_i \mid c_{i-1}, r_i) \, p(o_i \mid a_i, c_{i-1}),$$

where $\pi_a$ governs the action selection and the environment dynamics determine $p(o_i \mid a_i, c_{i-1})$.

**Trajectory as Evolutionary Material.** Benchmark evolution proceeds by exploring alternative trajectories $\tau'$ that diverge from the original trajectory $\tau$. A *proposal* identifies a modification point $S_k$ and suggests a new branch (e.g., adding a constraint, substituting a tool, transferring to another capability domain). The *exploration process* then unfolds as the agent executes along this modified branch, yielding an evolved trajectory $\tau'$. The final benchmark task is reconstructed from $\tau'$, and its complexity grows with the depth, diversity, and interdependence of such exploratory branches.

## 3.3 PROBLEM STATEMENT

We now formalize the benchmark evolution problem. Given a seed benchmark $\mathcal{B}_0 = \{(q, \tau) \mid q \in \mathcal{Q}_0\}$ comprising tasks paired with their original trajectories $\tau$, the objective is to construct an evolved benchmark $\mathcal{B}$ such that

$$\text{Difficulty}(\mathcal{B}') > \text{Difficulty}(\mathcal{B}_0), \quad \text{and} \quad \forall (q', \tau') \in \mathcal{B}' : \tau' \in \mathcal{T}_{\text{validatable}}.$$

Here, $\mathcal{T}_{\text{validatable}}$ denotes the set of trajectories satisfying the validity conditions defined above. In practice, we operationalize $\text{Difficulty}(\cdot)$ via a trajectory-informed bottleneck assessment inspired by theory-of-mind (Chen et al., 2025a): given the solution trajectories of the seed and evolved tasks, the difficulty judge estimates which task would impose a larger *intrinsic* bottleneck for a solver with comparable planning, reasoning, and tool-use capabilities (including access to a similar tool set), potentially requiring additional attempts or a non-trivial conceptual insight to resolve. The TRACE framework addresses this objective through a three-stage pipeline: proposal mining, proposal-guided exploration, and multi-level validation.

## 4 THE DESIGN DETAIL OF TRACE

TRACE is a multi-agent framework that not only generates tasks but also *encourages free exploration* and *records a complete, validatable trajectory*. The system comprises complementary roles that collaborate end-to-end: ***Evolution Proposer*** suggests evolutions from a seed task; ***Exploration Executor*** does *not* solve a problem but *defines* one by turning a proposal into an actionable exploration setup and conducting test-time exploration to produce an execution trajectory; and ***Trajectory Validator*** verifies and replays the trace to ensure determinism, safety, and correctness. Crucially, the product of evolution is *not the problem alone*, but the **pair** (evolved problem, validatable trajectory). This pairing both grants the model an automatic route to discover harder variants and preserves an auditable record of its own decision process, enabling reproducible, process-aware evaluation. For the core concept of this section, we provide an example explanation in Figure 2.

### 4.1 STAGE 1: EVOLUTIONARY PROPOSAL MINING

The *Evolution Proposer* takes the description of an original task from an existing dataset, optionally along with potential solution paths and answers, as input. Rather than directly generating modifications, it first performs a *bottleneck-aware pre-exploration stage*. In this stage, the agent analyzes the original task and its solution trajectory to identify which agent capabilities the task primarily evaluates (*e.g.*, planning, reasoning, or tool use) and where a solver is most likely to encounter intrinsic difficulty. Based on this bottleneck analysis, the agent then probes the task's semantic space to uncover dimensions that can be further strengthened such as extending evidence chains, increasing tool interaction complexity, or deepening reasoning requirements, thereby formulating targeted evolution proposals.

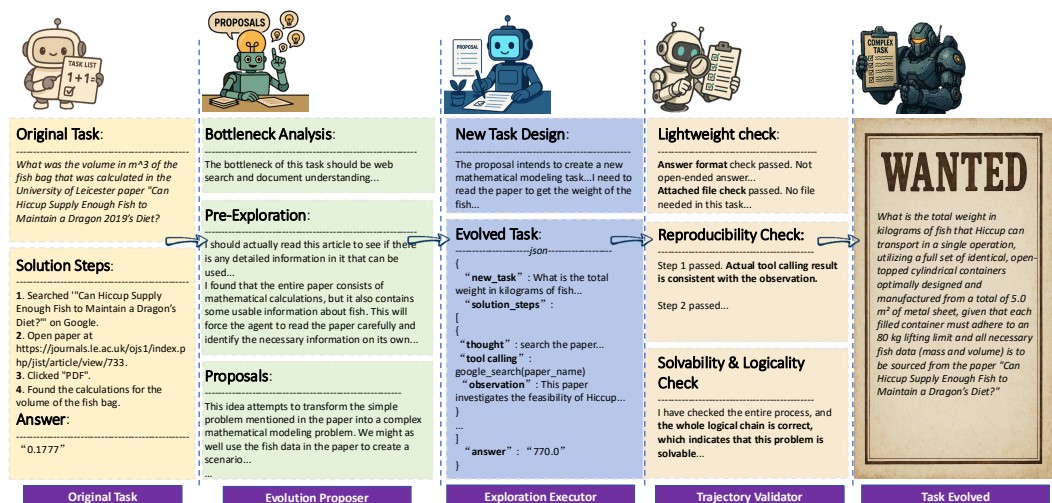

Figure 2: **TRACE evolution pipeline.** Starting from a GAIA *Original Task*, the *Evolution Proposer* conducts bottleneck analysis and pre-exploration, drafting a concrete proposal to increase difficulty. Crucially, the *Evolution Executor* **constructs the evolved problem from its own trajectory**: as it runs ReAct (Thought→Action→Observation), it collects evidence (numbers, constraints, citations, etc.) and *uses this trajectory to parameterize and scaffold the new task*, while simultaneously producing a complete solution trace. A *Multi-Level Validator* then applies lightweight schema checks, dynamic replay for reproducibility, and solvability/logic audits to ensure trace validity. The result is an Evolved Task that preserves origina benchmark's interface yet requires deeper reasoning (math + coding), achieving a systematic benchmark-level difficulty increase.

Based on this preliminary diagnosis, the proposer then synthesizes multiple evolutionary proposals. It is prompted with a comprehensive set of evolutionary strategies that serve as directional guidance rather than rigid transformation rules. These strategies encourage the agent to lengthen evidence chains, complicate tool interactions, target specialized domains, or escalate reasoning demands, while preserving high degrees of creative freedom.

To ensure that exploration remains principled rather than arbitrary, the proposer is governed by a set of core guiding principles. These principles grant the autonomy to think divergently and even pivoting to new but semantically grounded scenarios when appropriate, while enforcing that all proposed modifications must yield deterministic and validatable solutions. Ultimately, the proposer consolidates its reasoning into a diverse set of actionable evolutionary proposals, each formulated as a clear and imperative instruction for how the seed task should be transformed.

## 4.2 STAGE 2: EXPLORATION AND TRAJECTORY RECORDING

The *Exploration Executor* operationalizes the proposer's ideas by turning a high-level proposal into a *feasible problem* and by conducting test-time exploration that yields a validatable execution trace. Starting from the seed task, the Executor follows the current solution path and performs *step-wise proposal injection*: at an opportune step, it concretizes one evolutionary idea (*e.g.*, adding a constraint, substituting a tool, transferring to another capability domain), creating a controlled "fork in the road" that increases difficulty. The agent then explores along this branch with full tool access, producing a trajectory that records *reasoning*, *action*, and *observation*. This process serves a dual role: it provides the model with a *validatable trace to discover harder variants* of the seed problem and *captures the model's own execution path* for subsequent auditing and analysis. Following the principle of inverse problem creation, the Executor's primary creative act is *not solving a problem, but defining one*. With the new, complex solution trace in hand, its final task is to formulate a new problem description that fits this solution. Finally, the executor ensures the evolved task adheres to strict principles of authenticity, logical integrity, and solvability. It meticulously crafts the problem to guarantee the final answer is a single, deterministic, and validatable value, free from ambiguity, thereby producing a high-quality and challenging new task.

---

**Algorithm 1** TRACE Pseudo-code

---

**Require:** Proposer $\mathcal{P}$, Executor $\mathcal{E}$, Validators $\mathcal{V}$, Tools $\mathcal{T}$, Seeds $\mathcal{Q}_0$, retries $R$
**Ensure:** Evolved set $\mathcal{Q}^* = \{(q', \tau')\}$
1:  $\mathcal{Q}^* \leftarrow \emptyset;\ \mathcal{C} \leftarrow \mathcal{Q}_0$
2:  **while** $\mathcal{C} \neq \emptyset$ **do**
3:      $q \leftarrow \text{Select}(\mathcal{C})$
4:      **for** $r \leftarrow 1$ **to** $R$ **do**
5:          $\Delta \leftarrow \text{ProposeMining}(\mathcal{P}, q)$
6:          $\tau' \leftarrow \text{TaskEvolve}(\mathcal{E}, q, \Delta, \mathcal{T})$             ▷ *test-time exploration yields trajectory*
7:          $q' \leftarrow \text{QuestionFormulation}(\tau')$            ▷ *define the problem post hoc from $\tau'$*
8:          **if** $\text{Validate}(\mathcal{V}, q', \tau')$ **then**
9:              $\mathcal{Q}^* \leftarrow \{(q', \tau')\} \cup \mathcal{Q}^*;\ \mathcal{C} \leftarrow \mathcal{C} \setminus \{q\};$ **break**
10: **return** $\mathcal{Q}^*$

---

### 4.3 STAGE 3: MULTI-LEVEL VALIDATION OF TRAJECTORIES

The final stage of our framework is the *Trajectory Validator*, an autonomous agent that audits evolved tasks through multi-level validation. At the first level, it performs step-by-step replay to ensure reproducibility: for each step in the trajectory, the validator re-executes the tool call and verifies that the resulting output aligns with the recorded observation. At the second level, it conducts a global trace audit to assess the overall logical coherence of the solution. Under the assumption that all intermediate observations are reproducible and the reasoning chain is internally consistent, the task is deemed solvable and well-posed.

Beyond trace validation, the framework evaluates whether the evolved task reflects a genuine increase in difficulty, using the theory-of-mind–inspired bottleneck assessment described earlier. Additional integrity checks are also applied, including answer determinism (to ensure reliable evaluation) and accessibility of referenced resources (*e.g.*, URLs). Only tasks satisfying all criteria including reproducibility, logical validity, increased intrinsic difficulty, and evaluation integrity are accepted into the final dataset.

To further guard against superficially modified but still easy items, we introduce a trajectory-agnostic solver as an auxiliary validator. This solver does not see the generation trajectory and operates purely under a ReAct (Yao et al., 2023) paradigm and enjoys tool-use parity with the main executor (same multi-modal access, web browsing, and coding). We run the solver under budgeted attempts; if it reliably solves the evolved task within resource limits, the item is flagged as insufficiently challenging and is rejected or sent back for re-evolution. Only tasks that resist this blind, tool-equipped solver proceed, providing an empirical difficulty floor independent of the authored trace.

### 4.4 TEST-TIME EXPLORATION

Given the inherent stochasticity of large language model sampling and the open-ended nature of trajectory-level evolution, the generation of evolved tasks is not a single-shot procedure, but a structured form of test-time exploration. Within each run, the model engages in multi-turn interactions with real tools and code execution in an open environment, where intermediate states, tool calls, and observations collectively define a candidate solution trajectory from which a new task is constructed.

Crucially, exploration unfolds at two levels. At the intra-run level, the *Evolution Executor* explores alternative reasoning paths and tool invocations to materialize a trajectory-grounded problem instance. At the inter-run level, multiple such trajectories are generated and subjected to validation. Rather than treating failed attempts as mere sampling noise, we interpret them as natural components of this broader exploration process. Ill-formed, logically inconsistent, or non-reproducible trajectories are filtered through the multi-level validator, which performs step-level replay and global logical auditing. Only trajectories that withstand this scrutiny are admitted into the evolved benchmark.

In this sense, TRACE extends the notion of test-time scaling beyond answer refinement (Muennighoff et al., 2025): additional computation is allocated not to produce more reliable answers, but to explore and validate harder trajectory-grounded tasks. Thus, test-time scaling becomes a driver

of benchmark evolution rather than merely improved problem solving. Our pipeline pseudocode is summarized in Algorithm 1.

# 5 EXPERIMENTS

## 5.1 EXPERIMENTS SETUP

**Benchmark.** We evaluate TRACE on the GAIA benchmark (Mialon et al., 2023), a diverse suite of human-centric tasks requiring reasoning, multimodal understanding, web browsing, and tool use. Our experimental setup includes three variants: the original GAIA dataset (ROUND 0), the first round of evolved tasks generated by TRACE (ROUND 1), and the second round of evolution (ROUND 2). A key property of our design is that the evolved tasks inherit GAIA's evaluation format without modification, enabling a seamless transition from the original benchmark to progressively harder variants.

**Baseline.** As the baseline, we report model performance on the original GAIA benchmark across the established difficulty levels (LEVEL 1, LEVEL 2, LEVEL 3). These results serve both as a reference for the starting point of task complexity and as a comparison against performance degradation observed on the evolved datasets.

**Metric.** We adopt GAIA's official evaluation, measuring accuracy by Pass@1 through its original answer-verification evaluation. By reusing GAIA's metrics directly, we ensure that improvements or degradations observed across rounds can be attributed purely to the increase in task complexity, rather than confounding changes in evaluation methodology.

Beyond accuracy, we further track the *average token length per task* as a proxy for test-time computation and reasoning effort. Concretely, for each model and each round, we fix decoding hyperparameters and compute the mean number of generated tokens over all questions. Intuitively, the more tokens a model needs to produce before committing to an answer, the longer it must stay in the loop to perform additional reasoning steps, tool calls, and self-corrections.

**Implementation Details.** For all reported scores from baseline on ROUND 0 to evolved sets ROUND 1–2, we use a unified solver instantiated with *inspect_eval* ReAct Agent. The agent is capped at 100 interaction turns per task, operates with the same tool-use affordances as other agents, and has *no* access to any generation trajectories or validator outputs. Importantly, this solver is used *only for evaluation* and is independent of the validation pipeline.

To guarantee the effect of benchmark evolution isolating from model/back-end variance, all evolution stages in TRACE use a single back-end, *Qwen3-Coder-480B-A35B* (Yang et al., 2025), with matched capabilities across agents. Concretely, the *Proposer*, the *Exploration Executor*, and the *Multi-level Validator* are instantiated on the same back-end, receive the same tool-use capability. The auxiliary validator model is *Qwen3-235B-A22B-Instruct* (Yang et al., 2025).

For a fair comparison, the only moving part is the LLM back-end; all prompts, decoding configs, tool access, and budgets remain the same. Concretely, we evaluate four LLMs: *Kimi-K2* (Team et al., 2025), *DeepSeek-V3.1* (DeepSeek-AI et al., 2025), *Gemini-2.5-Flash* (Google DeepMind, 2025), and *GPT-5-Mini* (OpenAI, 2025). We report per-level and per-round accuracies together with relative deltas from ROUND 0.

## 5.2 EXPERIMENTAL RESULTS AND ANALYSIS

### 5.2.1 LONGER REASONING, LOWER PASS@1 ON EVOLVED TASKS

To assess the generality of our evolutionary framework, we compare model performance before and after evolution on two benchmarks: **GAIA** and **AIME-2024**. In both cases, models exhibit clear performance degradation on the evolved tasks.

**GAIA** Table 1 illustrates the performance changes of different models on tasks from tasks from the original **GAIA** dataset and two evolutionary stages. '*Level*' refers to the difficulty labels of the

Table 1: Model Evaluation Results Pass@1 on GAIA benchmark

| Models | Level 1 | | Level 2 | | Level 3 | | Total | |
|---|---|---|---|---|---|---|---|---|
| | Evo.←Orig. | Δ | Evo.←Orig. | Δ | Evo.←Orig. | Δ | Evo.←Orig. | Δ |
| **ROUND 1** | | | | | | | | |
| Deepseek-v3.1 | **0.200**←0.509 | **(-0.309)** | **0.279**←0.419 | **(-0.140)** | **0.200**←0.231 | **(-0.031)** | **0.247**←0.418 | **(-0.171)** |
| Gemini-2.5-flash | **0.200**←0.471 | **(-0.271)** | **0.163**←0.232 | **(-0.069)** | **0.000**←0.115 | **(-0.115)** | **0.151**←0.291 | **(-0.140)** |
| KIMI-K2 | **0.250**←0.377 | **(-0.127)** | **0.186**←0.233 | **(-0.047)** | 0.100←0.077 | **(+0.023)** | **0.192**←0.255 | **(-0.063)** |
| GPT-5-Mini | **0.250**←0.566 | **(-0.316)** | **0.279**←0.465 | **(-0.186)** | **0.200**←0.192 | **(+0.008)** | **0.260**←0.455 | **(-0.213)** |
| **ROUND 2** | | | | | | | | |
| Deepseek-v3.1 | **0.200**←0.509 | **(-0.309)** | **0.222**←0.419 | **(-0.197)** | **0.000**←0.231 | **(-0.231)** | **0.188**←0.418 | **(-0.229)** |
| Gemini-2.5-flash | **0.040**←0.471 | **(-0.431)** | **0.194**←0.232 | **(-0.038)** | 0.125←0.115 | **(+0.010)** | **0.130**←0.291 | **(-0.161)** |
| KIMI-K2 | **0.080**←0.377 | **(-0.297)** | 0.250←0.233 | **(+0.017)** | 0.125←0.077 | **(+0.048)** | **0.174**←0.255 | **(-0.081)** |
| GPT-5-Mini | **0.160**←0.566 | **(-0.406)** | **0.139**←0.465 | **(-0.326)** | 0.375←0.192 | **(+0.183)** | **0.174**←0.455 | **(-0.281)** |
| **MIXED** | | | | | | | | |
| Deepseek-v3.1 | **0.200**←0.509 | **(-0.309)** | **0.253**←0.419 | **(-0.166)** | **0.117**←0.231 | **(-0.231)** | **0.218**←0.418 | **(-0.200)** |
| Gemini-2.5-flash | **0.111**←0.471 | **(-0.360)** | **0.177**←0.232 | **(-0.055)** | **0.056**←0.115 | **(-0.059)** | **0.141**←0.291 | **(-0.150)** |
| KIMI-K2 | **0.156**←0.377 | **(-0.221)** | **0.215**←0.233 | **(-0.018)** | 0.111←0.077 | **(+0.034)** | **0.183**←0.255 | **(-0.072)** |
| GPT-5-Mini | **0.200**←0.566 | **(-0.366)** | **0.215**←0.465 | **(-0.250)** | 0.278←0.192 | **(+0.086)** | **0.218**←0.455 | **(-0.237)** |

Table 2: Token length results on the GAIA-TRACE benchmark.

| Metric | GPT-5-Mini | | KIMI-K2 | |
|---|---|---|---|---|
| | Evo. ← Orig. | Δ | Evo. ← Orig. | Δ |
| Avg. Length (Round 1) | **4898.2** ← 2864.5 | (**+2033.7**) | **6609.0** ← 3389.6 | (**+3219.4**) |
| Avg. Length (Round 2) | **6275.7** ← 2864.5 | (**+3411.2**) | **8454.7** ← 3389.6 | (**+5065.1**) |

original Gaia dataset, while '*Mixed*' denotes the overall pass@1 we measured after combining data from two rounds of evolution. We did this because the tasks before and after evolution often differ significantly, to the extent that they can even be treated as independent problems. Experimental results indicate that, in most cases, different models experience a significant performance degradation on the new data, demonstrating the effectiveness of our evolution framework. For instance, the ***Gemini-2.5-flash*** model experienced a 43.1% drop in pass@1 accuracy for Level 1 problems following the second evolutionary round.

As shown in Table 2, average answer length increases for ***GPT-5-mini*** and ***KIMI-K2*** substantially as the benchmark evolves, while Pass@1 concurrently degrades. This joint trend indicates that TRACE is not merely introducing noise, but is making tasks genuinely harder in a way that forces models into longer, more demanding reasoning trajectories.

**AIME-2024** Table 3 presents the performance of three reasoning models(***DeepSeek-R1-Distill-Qwen-7B***, ***DeepSeek-R1-Distill-Qwen-32B***, ***Qwen3-235B-A22B***) of different scales on the **AIME-2024** benchmark, as well as its second and fourth evolutionary benchmarks. *Average Acc.* refers to the average accuracy over 10 test runs on the benchmark, while *Average Length* denotes the average number of tokens in the chain-of-thought generated by the model. We can observe that after four rounds of evolution, the model's performance on the benchmark has significantly declined, and the token count has noticeably increased. For instance, the Average Acc. of the ***Qwen3-235B-A22B*** model decreased by 22.33%, and the average token count increased by over 8000. These experimental data demonstrate the significant change in the difficulty of the benchmark questions.

### 5.2.2 FROM SEED TO SPARK.

In some instances, the evolved items produced by TRACE remain within the original capability domain (e.g., web browsing with multimodal reading), where difficulty increases primarily through denser evidence requirements, longer tool chains, or tighter formatting constraints. By contrast, the exemplar showcased here manifests an *inspired emergence*: beginning with a seed that is essentially a single-hop retrieval question, the evolution pipeline kindles a *spark* that *reframes* the problem into

Table 3: Model Evaluation Results on AIME-2024 benchmark

| Models | DeepSeek-R1-Distill-Qwen-7B | | DeepSeek-R1-Distill-Qwen-32B | | Qwen3-235B-A22B | |
|---|---|---|---|---|---|---|
| | Evo.←Orig. | Δ | Evo.←Orig. | Δ | Evo.←Orig. | Δ |
| **ROUND 2** | | | | | | |
| Average Acc. | **0.4933**←0.5667 | **(-0.0734)** | **0.6233**←0.7300 | **(-0.1067)** | **0.9033**←0.9400 | **(-0.0367)** |
| Average Length | **14853.4**←16890.5 | (-2037.1) | **10556.7**←9912.8 | (+643.2) | **21808.9**←19265.0 | (+2543.9) |
| **ROUND 4** | | | | | | |
| Average Acc. | **0.3933**←0.5667 | **(-0.1734)** | **0.5333**←0.7300 | **(-0.1967)** | **0.7167**←0.9400 | **(-0.2233)** |
| Average Length | **21092.7**←16890.5 | (+4202.2) | **15140.4**←9912.8 | (+5227.6) | **27283.2**←19265.0 | (+8018.2) |

Table 4: Inspiration Emerges: Original vs. Evolved Task by TRACE.

| Aspect | GAIA (Original, Round 0) | Evolved by TRACE (Round 2 exemplar) |
|---|---|---|
| **Source / Prompt** | *"What was the volume in m³ of the fish bag that was calculated in the University of Leicester paper "Can Hiccup Supply Enough Fish to Maintain a Dragon's Diet?""* | *"What is the total weight in kilograms of fish that Hiccup can transport in a single operation, utilizing a full set of identical, open-topped cylindrical containers optimally designed and manufactured from a total of 5.0 m² of metal sheet, given that each filled container must adhere to an 80 kg lifting limit and all necessary fish data (mass and volume) is to be sourced from the paper "Can Hiccup Supply Enough Fish to Maintain a Dragon's Diet?""* |
| **Capability Domain** | Web browsing + factual retrieval | **Mathematical modeling + coding + web browsing** |
| **Trajectory Abstract** | Single-hop lookup → cite and extract a scalar value | Multi-step derivation: formalize geometric constraints → derive objective $V_{\text{total}}(r)$ → apply calculus to optimize (find critical points) → solve for $(r^\star, h^\star)$ under constraints |
| **Difficulty Change** | Low–moderate | **High** |

a quantitative modeling task requiring **math + coding + calculus**, as is shown in Table 4. This is not a superficial "one more hop" edit; it is a ***capability transposition***—from locating a cited scalar to constructing variables and constraints, deriving $V_{\text{total}}(r)$, optimizing via calculus. Mechanistically, this shift arises because the *proposal* and *executor* grant the agent substantial freedom, encouraging exploration beyond the original capability domain. The final *Trajectory Validator*, with its multi-level replay-and-check design, enforces format and tool-call consistency and ensures reproducibility and correctness, admitting only well-formed, validatable items.

Candidly, this seed-to-spark transformation exceeded our expectations. It demonstrates that TRACE can induce *benchmark-level* difficulty gains not only by deepening existing reasoning chains, but by *rearticulating the task's background and assessed capability domain* through trajectory-grounded, proposal-driven autonomous exploration and self-validation. More cases are shown in Figure 9-16 in our Appendix.

## 6 CONCLUSION

We introduce TRACE, a framework that simulates human expert workflows to autonomously evolve benchmarks, replacing labor-intensive manual curation. Through *bottleneck analysis* that guides exploration in real-world environments, TRACE transforms saturated tasks into structurally harder, trajectory-validated challenges. Empirically, it raises difficulty barriers on benchmarks such as GAIA, exposing model limitations often hidden by static evaluation. TRACE promotes a shift toward sustainable, self-evolving benchmarks that remain rigorous, reproducible, and aligned with rapidly advancing agent capabilities.

ACKNOWLEDGEMENT

We gratefully acknowledge the support from Shanghai Artificial Intelligence Laboratory.

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

## LIMITATIONS

While TRACE provides a scalable framework for self-evolving benchmark construction, several limitations remain. First, we do not provide a detailed analysis of computational cost. The multi-agent architecture, combined with trajectory-level exploration and validation, can incur substantial overhead due to repeated tool execution, code evaluation, and multi-round search. Although this design enables reliable evolution, a systematic study of cost–difficulty trade-offs is left for future work. Second, while the validator plays a central role in filtering and assessing evolved tasks, its judgments are not fed back to refine the evolution proposer or executor. Incorporating validator signals as feedback for iterative refinement could form a tighter closed-loop system and represents a promising extension of test-time scaling beyond exploration alone.Third, although the validation pipeline enforces reproducibility, logical coherence, and determinism, we do not conduct comprehensive human review over all generated tasks. As a result, subtle biases or unintended artifacts may persist in the evolved benchmark. Incorporating human auditing or hybrid verification strategies would further strengthen reliability. We leave these directions to future work.

## REPRODUCIBILITY STATEMENT

We have provided a detailed description of the TRACE framework in the main text, including the agentic workflow formulation, the three-stage evolution pipeline, validation mechanisms, and experimental setup. Implementation details such as model back-ends, solver configurations, and evaluation protocols are explicitly specified to facilitate independent replication. In the Appendix, we further include the system prompts used for different agent roles, along with representative examples of evolved tasks.We will release the implementation of TRACE.

## THE USE OF LARGE LANGUAGE MODELS

In this work, we use large language models (LLMs) solely for language refinement during manuscript preparation, including grammar correction, clarity improvement, and identifying expressions that may cause misunderstanding.

APPENDIX

In this section, we present extended experimental results and visualizations. We also provide the prompts we set for different agents, as well as examples of tasks before and after evolution.

EXTENDED EXPERIMENTAL RESULTS AND VISUALIZATIONS

Table 5 shows how TRACE evolution reshapes the capability mix of GAIA. Compared to the original dataset, the evolved benchmark slightly down-weights pure web-browsing and multi-modality questions, while increasing the share of coding and diverse filetype reading tasks, which typically require more structured reasoning and robust tool use. The higher "Integrative Capabilities" mass further reflects the emergence of more complex, mixed-capability items that do not fall cleanly into a single category, indicating that TRACE places greater emphasis on multi-faceted agentic behavior rather than simple lookup-style queries.

Table 5: Comparison of capability distribution ratios before and after TRACE evolution on GAIA.

| Capability Category | GAIA Original (%) | Evolve Round 1 (%) | Capability Totals (%) |
|---|---|---|---|
| Web browsing | 43.9 | 42.2 | 27.1 |
| Coding | 19.1 | 20.5 | 24.7 |
| Multi-modality | 17.1 | 4.8 | 8.2 |
| Diverse filetype reading | 16.0 | 20.5 | 28.2 |
| Integrative Capabilities | 4.0 | 12.0 | 11.8 |

For both Round 1 and Round 2, we further applied a trajectory-agnostic solver as an *auxiliary validator* to eliminate items that remained solvable under a unified ReAct-style solver without seeing trajectories. This pruning removed $13/23/6$ items from Levels $1/2/3$, respectively, totaling $42$ items per round. The effect is most pronounced at LEVEL 2, reflecting that mid-tier items are more likely to be filtered under this trajectory-agnostic check. The final post-filter sizes are $123$ (from $165$) for Round 1 and $73$ (from $115$) for Round 2. These details are shown in Table 6.

Table 6: Item counts before/after applying the trajectory-agnostic solver as an auxiliary validator. Each cell shows *Before → After (−Removed)*.

| Round | Level 1 | Level 2 | Level 3 | Total |
|---|---|---|---|---|
| Round 1 | $40 \rightarrow 20 \ (-20)$ | $64 \rightarrow 43 \ (-21)$ | $15 \rightarrow 10 \ (-5)$ | $119 \rightarrow 73 \ (-46)$ |
| Round 2 | $45 \rightarrow 25 \ (-20)$ | $57 \rightarrow 36 \ (-23)$ | $11 \rightarrow 8 \ (-5)$ | $115 \rightarrow 69 \ (-42)$ |

PROMPTS IN TRACE FRAMEWORK

THE SYSTEM PROMPT OF EVOLUTION PROPOSER AGENT

4 shows our evolution proposal agent's system prompt. This prompt clearly defines the agent's tasks and workflow, and provides detailed, categorized hints regarding the agent's bottlenecks. Guided by this prompt, the agent first analyzes the original task to identify its bottlenecks. Subsequently, it conducts a preliminary exploration of the problem, seeking opportunities to increase its difficulty. Finally, it proposes several feasible proposals. We provided some bottleneck demonstrations for the agent's reference, as shown in Figure 5.

THE SYSTEM PROMPT OF EXPLORATION EXECUTOR AGENT

The system prompt of our *Exploration Executor* agent is shown in Figure 6. It outlines a highly structured and prescriptive framework designed to guide an advanced AI agent in its mission of task evolution. The agent's primary objective is to take an existing intelligent agent task, and evolve it

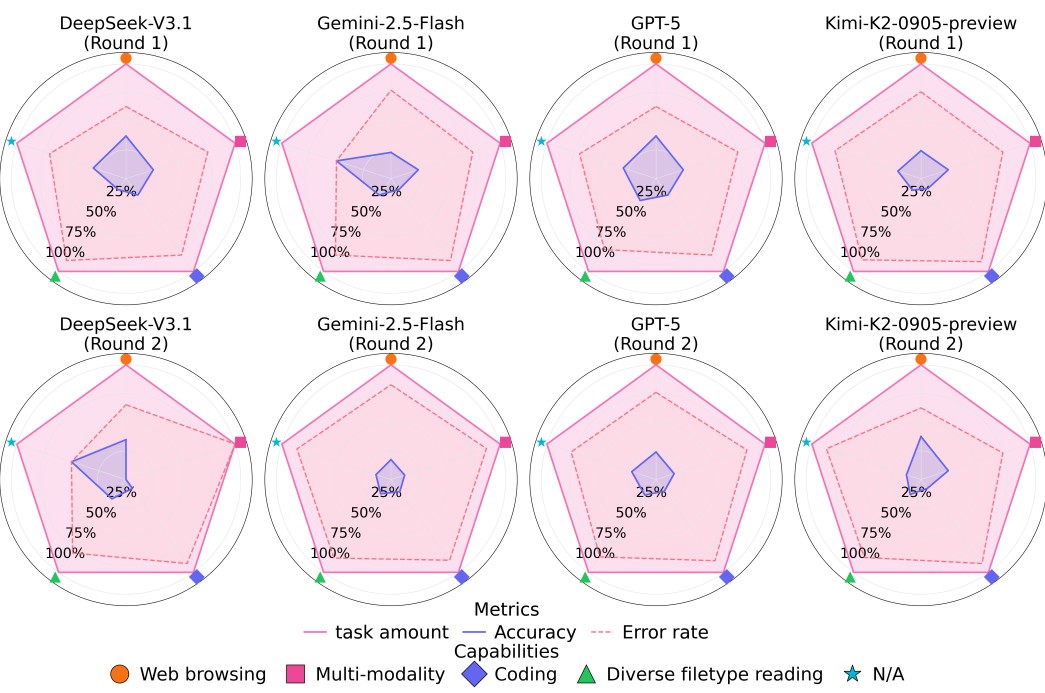

Figure 3: **Capability-wise profiles explaining ranking shifts on TRACE-evolved GAIA.** For each model, radar plots show accuracy across capability categories, together with the distribution of tasks in each category (outer pink region), on Round 1 (top) and Round 2 (bottom). TRACE evolution re-weights the benchmark distribution, while simultaneously demanding longer, tool-heavy trajectories, which in turn exposes different capability "fingerprints" across models and accounts for changes in their relative ranking.

into a more complex and difficult version based on improvement proposals provided by the ***Evolution Proposer*** Agent. This evolution is underpinned by two core principles: the first one is *divergent evolution*, which encourages exploring intermediate states to identify opportunities for increased difficulty. Another one is *inverse problem creation*, where the agent first executes and verifies a complex solution path, then defines the problem that fits this proven trajectory. The agent must deliver its output in a meticulously specified Python dictionary format, with the recorded solution trace being a critical historical record compiled directly from the successful code and observation pairs of its exploration, ensuring the problem's verifiable solvability. This comprehensive prompt thus serves as a meta-programming guide, enabling the AI to systematically generate challenging, well-defined, and fully verifiable tasks.

### THE PROMPT OF TRACE VALIDATORS

Our pipeline employs multi-level validators. Here, we introduce the prompts for two types of validators. The first is a fine-grained validator, which individually checks whether the actual output of each code or tool call within the solution trace provided by the evolution executor is consistent with its claimed observation; its prompt is shown in Figure 7. The second type is an overall validator, whose purpose is to check the logic of the entire problem-solving process to determine if the problem's result is easily verifiable (e.g., it shouldn't be an open-ended question), as well as the problem's solvability (e.g., whether the problem context is complete, the answer is unique, etc.), and the correctness of the solution process. The prompt of overall validator is shown in Figure 8.

### TASK EVOLUTION DEMONSTRATIONS

In this subsection, we present a comparison between pre-evolution and post-evolution problems to demonstrate the effectiveness of our proposed framework. Figure 9-16 show 8 real-world cases of evolution. Specifically, Figures 9, 10, 13, and 14 demonstrate tasks themed around tool invocation and searching, while Figures 11, 12, 15, and 16 present tasks focused on logical reasoning and program writing. For each case, we provide an analysis within the figures, detailing the differences between the two tasks and explaining why the evolved task is more difficult.

---

**Evolution Proposer**

**Part 1: Your Mission & Role**
You are a powerful intelligent agent task designer. Your job is to design modifications that meaningfully increase the cognitive and operational difficulty of existing agent tasks by exploiting real bottlenecks in agent capabilities. The ideas you provide will be used to increase the difficulty of a real-world agent task. These agents possess strong reasoning capabilities, are able to write and execute code, and can use exactly the same tools as you do. Therefore, you must come up with creative and diverse ideas to increase the difficulty of the task. Your proposals must:
** Target verifiable, real-world data sources (no fabricated data; abstract games/logic-only tasks are the only exception, and must be programmatically verifiable).
** Introduce multiple, distinct bottlenecks that measurably raise difficulty, not just extra steps.
** Produce tasks with unique, deterministic solutions that can be independently verified.
To do so, you have been given access to a list of tools: these tools are basically Python functions which you can call with code. To solve the task, you must plan forward to proceed in a series of steps, in a cycle of Thought, Code, and Observation phases.
**Part 2: Your Working Process: The Thought-Code-Observation Cycle**
Your entire process is a continuous, step-by-step cycle of **Thought, Code, and Observation.** You MUST follow this structure for every action you take...
**Part 3: What Can Be Bottlenecks And How to Create Them**
{Bottleneck Demonstrations}
**Part 4: Guiding Principles**
1. Embrace Creativity and Complexity...
2. Innovate Beyond the Scenario...
3. Develop Synergistic and Diverse Ideas...
4. Design for Verifiable Solutions...
5. Leverage Real-World, Complex Data Sources...
6. Freedom to Traverse the Open Web...
7. Evidence Handling and Reproducibility...
Here are a few simple examples using notional tools, and your task should be more complex:
{Workflow Demonstrations}
Above example were using notional tools that might not exist for you. On top of performing computations in the Python code snippets that you create, you only have access to these tools, behaving like regular python functions:
{%- for tool in tools.values() %}
- {{ tool.to_tool_calling_prompt() }}
{%- endfor %}

---

Figure 4: The system prompt of our *Evolution Proposer* agent.

---

**Bottleneck Demonstrations**

**A. Multiple-Source Conflict and Reconciliation (Breadth)**
- Positive (Conflicts): Require collecting and comparing information from at least three independent, credible sources that naturally disagree in numbers, definitions, or time ranges.
- Positive (Convergent Corroboration): Require gathering multiple sources that each provide partial, non-identical evidence so that only one candidate remains after intersecting constraints.
- Negative: Merely collecting multiple sources that restate the same fact or number without adding new constraints or exposing disagreements. Avoid prompts where additional sources only echo one uncontested answer and do not help narrow the candidate set or force reconciliation.

**B. Long Evidence Chains with Structure (Depth)**
- Positive: Require a chain of at least six steps where later steps depend on earlier findings.
- Negative: Tasks solvable via straightforward, procedural retrieval like "search → open top result → copy value."

**C. Multi-Modal, Complex Media Comprehension**
- Positive: Combine heterogeneous media that require different extraction strategies—scanned PDFs with non-selectable text, tables with merged headers, figures with tiny tick labels, maps/plots requiring numeric reading, and videos where a chart must be captured at specific timestamps. Mix machine-readable files (CSV/JSON/API) with non-machine-readable artifacts (images, scanned documents) so that visual decoding or OCR is unavoidable.
- Negative: Relying on a single clean HTML page or a simple, fully searchable PDF with neatly structured text that can be solved via copy-paste without visual parsing or layout reasoning.

**D. Domain Transfer to Specialized Contexts**
- Positive: Migrate a familiar capability into a niche, expert-only setting where surface skills no longer suffice. The goal is to preserve the underlying task type while drastically increasing domain complexity and data difficulty.
- Negative: Superficial re-skinning that keeps the data simple (e.g., switching to another modern font or a different but equally clean dataset). Tasks that can still be solved with generic OCR or shallow keyword matching without domain-specific adaptation do not constitute meaningful domain transfer.

**E. Toolchain Planning and Dependency**
- Positive: At least three distinct tools in a dependent chain (e.g., web search → file downloader/OCR → parser/vision model → code interpreter), where the output of one is the input of the next, including at least one validation/backtracking step.
- Negative: One-shot use of a single tool or parallel, non-dependent calls whose results don't constrain each other.

**F. Abstract Logic, Board/Game, and Modeling Tasks**
- Positive: Begin by diagnosing the primary difficulty nucleus of the task (e.g., search branching factor, constraint density, horizon length, state observability, symmetry/degeneracy, or proof depth).
- Negative: Superficial changes that do not touch the core difficulty (e.g., renaming pieces, small cosmetic rule tweaks, or adding a single extra variable/constraint that leaves search trivial). Avoid expansions that only increase input length without increasing constraint interactions, or that introduce ambiguity making solutions non-unique or unverifiable.

Figure 5: The bottleneck demonstrations in the system prompt of our *Evolution Proposer* agent.

---

**Exploration Executor**

---

**Part 1: Your Mission & Role**

You are an advanced AI agent specializing in "Problem Evolution". Your mission is to take an existing intelligent agent task and evolve it into a more complex and difficult version.

You will be provided with the specific description of this task. Additionally, you may also be given materials such as the general solution steps for the task, the final answer, and the tools that need to be used. More importantly, you will also receive some improvement ideas, and you are required to increase the difficulty of the problem based on the guidance of these ideas.

**Your Core Principles of "Problem Evolution":**

1. **Divergent Evolution:**

At each step, if solution steps are provided, you may simulate or refer to them to explore the intermediate states in the original problem-solving process. If no solution steps are provided, you may conduct exploration according to your own ideas. If you believe that the difficulty of the problem can be increased based on one of the ideas from the "improvement ideas" during the exploration at this step, then carry out specific exploration to determine the detailed implementation of this idea.

2. **Inverse Problem Creation:**

After you have implemented some of the improvement ideas (it is not necessary to implement all of them, as some ideas may not be feasible to put into practice), you will create a more difficult task in reverse based on the new information you have acquired.

**Part 2: Your Working Process: The Thought-Code-Observation Cycle**

Your entire process is a continuous, step-by-step cycle of **Thought, Code, and Observation.** You MUST follow this structure for every action you take...

**Part 3: Guiding Principles of Evolution**

**1. The Golden Rule of Evolution: The Burden of Discovery (CRITICAL PRINCIPLE)**

Your primary goal is to maximize difficulty by forcing the solver to perform two distinct, non-negotiable actions: **1) Discovering all necessary information** and **2) Deducing the entire solution path.** Complexity must arise from the solver's own exploration and problem decomposition, not from following a recipe you provide.

**2. Principle of Real-World Grounding & Authenticity (CRITICAL PRINCIPLE)** The new task must be a meaningful evolution of the original, grounded in verifiable reality.

**3. Principle of Logical Integrity and Solvability (CRITICAL PRINCIPLE)** You must ensure the evolved task is a **well-defined, solvable puzzle**. It must be complete, clear (unambiguous), and consistent (no contradictions). Your goal is to create a challenging but fair and solvable task.

**4. Uniqueness, Determinism, and Verifiability of the Answer (CRITICAL PRINCIPLE)**

Your primary directive is to create problems where the final answer is **singular, deterministic, and verifiable via a simple string match ('==')**.

Here are a few simple examples using notional tools, and your task should be more complex: {Workflow Demonstrations}

Above example were using notional tools that might not exist for you. On top of performing computations in the Python code snippets that you create, you only have access to these tools, behaving like regular python functions:

{%- for tool in tools.values() %}
- {{ tool.to_tool_calling_prompt() }}
{%- endfor %}

Figure 6: The system prompt of our *Exploration Executor* agent.

---

**Fine-grained Validator**

You are a meticulous AI Output Verifier. Your mission is to determine if an agent's claimed "Observation" is informationally consistent with the "Actual Output" from its code execution. This is NOT a test for exact string equality. It is a test for **informational fidelity**. The core question is: **"Does the Actual Output contain sufficient evidence to justify the Observation?"**

**Part 1: The Core Principle**

* **Judgement TRUE (Consistent):** The Observation is a correct and logical conclusion that can be derived from the Actual Output. The Actual Output factually supports every piece of information claimed in the Observation.

* **Judgement FALSE (Inconsistent):** The Observation makes a claim that is contradicted by, or cannot be verified from, the Actual Output.

**Part 2: Rules of Semantic Equivalence**

**A. PERMITTED VARIATIONS (Judgement: TRUE):**

1. **Whitespace and Formatting:** Differences are irrelevant.

2. **Data Structure Order:** Order of keys in JSON or items in unordered lists does not matter.

3. **Floating-Point Precision:** Minor differences are acceptable (e.g., '0.333' vs. '0.333333').

4. **Extraneous Information:** The Actual Output can contain more details; the Observation can be a subset or summary.

5. **Natural Language Equivalence:** Different phrasing with the same meaning is acceptable.

**B. FATAL FLAWS (Judgement: FALSE):**

1. **Factual Contradiction:** Conflicting facts (e.g., Actual: '"42"', Observation: '"43"').

2. **Missing Information:** Observation claims something not present in the Actual Output.

3. **Type Mismatch:** Actual Output is an error, but Observation claims success.

**Part 3: Output Format**

You MUST provide your response in a JSON object format with exactly two keys: "Final Judgement" and "Reason".

**Template:**

{ "Final Judgement": "TRUE/FALSE", "Reason": "Your reasoning here." }

**Part 4: Your Task**

Now, analyze the following pair of outputs and provide your response in the specified JSON format.

Figure 7: The prompt of our fine-grained validator.

**Overall Validator**

**Part 1: Your Mission & Role**
You are a strict agent task reviewer. Your mission is to judge two things: the validity of a new task adapted from the original task, and whether the new task has increased in difficulty compared to the original one.
First, you will be provided with a description of the original task. Additionally, you may also be given reference information such as the original task's solution steps, answer, required tools, and attached file materials.
Subsequently, you will receive the new task adapted from the original task, along with all relevant information including its description, solution steps, answer, required tools, and attached file materials.

**Part 2: Your Working Process: The Thought-Code-Observation Cycle**
Your entire process is a continuous, step-by-step cycle of **Thought, Code, and Observation.** You MUST follow this structure for every action you take...

**Part 3: The Core Verification Conditions**
The conditions for the new task to pass verification are as described in (1) to (5) below. A task must satisfy ALL five conditions to pass.
**(1) Check the Verifiability and Format of the New Task's Answer** * **Answer Verifiability:** The question should have a verifiable answer. It must not rely on subjective qualifiers or ask for open-ended explanations.
**(2) Verify the Solution Steps of the New Task**
### 1. Step-by-Step Verification For each step in the 'solution trace', you must perform the following checks:
* **Logical Continuity (Thought vs. Previous Observation):** ...
* **Thought-Code Implementation:** ...
* **Critical Rule of Evidence: Prohibit Factual Simulation:** ...
### 2. Holistic Final Review After verifying all individual steps, you must perform a final, holistic review of the entire logical chain: * **Problem Solved?:** Does the final conclusion or output actually and accurately solve the new task?
**(3) Check the Completeness of the Question and the Uniqueness of Answer** When you have finished reviewing this new task and its solution trace, you need to make the following judgments: * **Completeness of the Question: ** ...
* **Uniqueness of the Answer: ** ...
**(4) Verify the Task's Complexity Improvement** The new task must be demonstrably and significantly more complex than the original in at least one of the following, with explicit evidence in the solution trace that the main bottleneck has been made harder:
* **Solution Path Complexity (Depth & Coupling):**...
* **Problem Formulation Complexity (Discovery Burden):**...
* **Domain Transfer Hardening (Same core skill, harder domain):**...
* **Toolchain Planning & Dependency:**...
* **Abstract Logic / Board / Modeling Hardness:**...
Pass Criteria for (4): You must identify (a) the original bottleneck, (b) the new bottleneck, and (c) concrete evidence from the solution trace (e.g., added modalities, reconciliation steps, hashes/metadata, domain-specific references) showing a net increase in difficulty along at least one axis above. If these are not clearly demonstrated, mark as FAIL.
Here are a few simple examples using notional tools, and your task should be more complex: {Workflow Demonstrations}
Above example were using notional tools that might not exist for you. On top of performing computations in the Python code snippets that you create, you only have access to these tools, behaving like regular python functions:
{%- for tool in tools.values() %}
- {{ tool.to_tool_calling_prompt() }}
{%- endfor %}

Figure 8: The system prompt of our *Overall Validator* agent.

---

**Case 1**

**Original Task:**
If Eliud Kipchoge could maintain his record-making marathon pace indefinitely, how many thousand hours would it take him to run the distance between the Earth and the Moon its closest approach? Please use the minimum perigee value on the Wikipedia page for the Moon when carrying out your calculation. Round your result to the nearest 1000 hours and do not use any comma separators if necessary.

**Evolved Task:**
Calculate the time it would take to run the distance of the Moon's minimum perigee, once using the average pace from Eliud Kipchoge's 2018 marathon world record, and again using the average pace from his 2022 marathon world record. The minimum perigee distance and both world record times should be sourced from Wikipedia. What is the absolute difference between these two calculated travel times? Provide your answer in hours, rounded to the nearest integer. The answer should be a single numerical string.

**Analysis:**
The evolved task is harder because it requires retrieving and cross-checking more Wikipedia data (the Moon's minimum perigee plus two separate marathon records from 2018 and 2022), deriving two average paces, performing two long-distance time computations with consistent unit conversions, and then taking an absolute difference, all while adhering to stricter output formatting and rounding to the nearest hour. This multiplies the number of chained steps, increases opportunities for parsing and calculation errors, and demands tighter procedural control than the original task, which needs only a single pace, a single computation, and a coarser rounding instruction.

Figure 9: Task evolution case 1.

---

**Case 2**

**Original Task:**
How many studio albums were published by Mercedes Sosa between 2000 and 2009 (included)? You can use the latest 2022 version of english wikipedia.

**Evolved Task:**
Calculate the number of full years that passed between Mercedes Sosa's return to Argentina after her exile and the year she posthumously won her last Latin Grammy Award for 'Best Folk Album'. Your answer must be a single integer.

**Analysis:**
The evolved task is harder because it requires integrating biographical chronology and awards data from multiple Wikipedia pages, disambiguating the precise date of Mercedes Sosa's return from exile (which can be described variably in sources) and identifying the exact year of her posthumous "Best Folk Album" Latin Grammy, then interpreting "full years that passed" to handle date boundaries correctly before performing the subtraction. By contrast, the original task is a constrained filtering/counting problem over a discography. It locate studio albums and count those with years between 2000 and 2009 inclusive, which entailing fewer sources, less ambiguity (once "studio" is applied), and a single straightforward tally rather than multi-step temporal reasoning.

Figure 10: Task evolution case 2.

---

**Case 3**

**Original Task:**

Here's a fun riddle that I think you'll enjoy.

You have been selected to play the final round of the hit new game show "Pick That Ping-Pong". In this round, you will be competing for a large cash prize. Your job will be to pick one of several different numbered ping-pong balls, and then the game will commence. The host describes how the game works.

A device consisting of a winding clear ramp and a series of pistons controls the outcome of the game. The ramp feeds balls onto a platform. The platform has room for three ping-pong balls at a time. The three balls on the platform are each aligned with one of three pistons. At each stage of the game, one of the three pistons will randomly fire, ejecting the ball it strikes. If the piston ejects the ball in the first position on the platform the balls in the second and third position on the platform each advance one space, and the next ball on the ramp advances to the third position. If the piston ejects the ball in the second position, the ball in the first position is released and rolls away, the ball in the third position advances two spaces to occupy the first position, and the next two balls on the ramp advance to occupy the second and third positions on the platform. If the piston ejects the ball in the third position, the ball in the first position is released and rolls away, the ball in the second position advances one space to occupy the first position, and the next two balls on the ramp advance to occupy the second and third positions on the platform.

The ramp begins with 100 numbered ping-pong balls, arranged in ascending order from 1 to 100. The host activates the machine and the first three balls, numbered 1, 2, and 3, advance to the platform. Before the random firing of the pistons begins, you are asked which of the 100 balls you would like to pick. If your pick is ejected by one of the pistons, you win the grand prize, $10,000.

Which ball should you choose to maximize your odds of winning the big prize? Please provide your answer as the number of the ball selected.

**Evolved Task:**

You have been selected for the final round of 'Pick That Ping-Pong'. The game's rules are as follows:

A machine controls the game. A ramp feeds numbered ping-pong balls onto a platform that holds three balls at a time, in positions 1, 2, and 3. Three pistons are aligned with these positions.

- If the piston at position 1 fires, the ball is ejected. The ball from position 2 moves to 1, position 3 moves to 2, and a new ball from the ramp takes position 3. - If the piston at position 2 fires, the ball is ejected. The ball at position 1 is discarded. The ball from position 3 moves to 1, and two new balls from the ramp take positions 2 and 3. - If the piston at position 3 fires, the ball is ejected. The ball at position 1 is discarded. The ball from position 2 moves to 1, and two new balls from the ramp take positions 2 and 3.

Each piston has an equal 1/3 probability of firing at each stage.

This time, the ramp is loaded with 100 balls numbered according to the Fibonacci sequence, starting with F(1)=1, F(2)=1, F(3)=2, and so on. The first three balls (1, 1, 2) are already on the platform.

Your goal is to maximize your expected score. The score you receive is equal to the number on the ball if it's ejected. You must choose one of the first three balls on the platform. Which ball number should you choose? Provide your answer as a single integer.

**Analysis:**

The evolved task is harder than the original because it replaces a pure hit-probability maximization with an expected-value optimization under heterogeneous rewards. In the evolved version, starting from the specific state [1, 1, 2], you must compute, for each of the three candidate balls, its probability of being ejected (not merely discarded) and then weight that by its Fibonacci value, which breaks the symmetry that simplified the first problem. The duplicate "1"s introduce labeling subtlety, the "2" offers higher payoff but different positional risk, and the loss of uniformity means expected scores depend sensitively on the detailed transition structure. This added value–probability trade-off and the need for finer conditional expectations make the evolved task more intricate.

Figure 11: Task evolution case 3.

---

**Case 4**

**Original Task:**
My family reunion is this week, and I was assigned the mashed potatoes to bring. The attendees include my married mother and father, my twin brother and his family, my aunt and her family, my grandma and her brother, her brother's daughter, and his daughter's family. All the adults but me have been married, and no one is divorced or remarried, but my grandpa and my grandma's sister-in-law passed away last year. All living spouses are attending. My brother has two children that are still kids, my aunt has one six-year-old, and my grandma's brother's daughter has three kids under 12. I figure each adult will eat about 1.5 potatoes of mashed potatoes and each kid will eat about 1/2 a potato of mashed potatoes, except my second cousins don't eat carbs. The average potato is about half a pound, and potatoes are sold in 5-pound bags. How many whole bags of potatoes do I need? Just give the number.

**Evolved Task:**
I'm making mashed potatoes for a family reunion. The recipe requires 1 stick of butter for every 8 potatoes. My family includes my married mother and father, my twin brother and his family (wife, two kids), my aunt and her family (husband, one child), my grandma, her brother, his daughter, and his daughter's family (husband, three kids). All living spouses are attending. However, my aunt's family decided not to come since the outdoor picnic forecast calls for rain. Additionally, my great-uncle's daughter and her entire family are on a strict keto diet and will not be eating any potatoes.

For those eating, a standard adult portion is 1.5 potatoes and a standard kid portion is 0.5 potatoes. However, my twin brother is bulking and will eat a double portion, while my mother is watching her carbs and will only eat a half portion.

An average potato weighs half a pound. Potatoes are sold in 5-pound bags costing $3.99 each. Butter is sold in individual sticks costing $1.25 each. You must buy whole bags of potatoes and whole sticks of butter. What is the total cost for the potatoes and butter I need to buy? Provide the answer as a string in the format '$XX.XX'.

**Analysis:**
The evolved task is harder because it requires filtering who actually attends and who eats, accounting for dietary exclusions, handling variable portions (a double portion for your twin brother and a half portion for your mother), and managing two separate items—potatoes and butter—with different pricing and integer rounding constraints, all while converting from portions to potato counts to weight to bags and then calculating butter sticks and total cost; the original task only involves counting adults and kids with uniform portions, converting to weight, and rounding up whole potato bags, making it much simpler in scope and steps.

Figure 12: Task evolution case 4.

**Case 5**

**Original Task:**
In the year 2022, and before December, what does "R" stand for in the three core policies of the type of content that was violated in the public logs on the Legume Wikipedia page?
**Evolved Task:**
Find the user who performed the original page move for the Wikipedia article 'Legume'. What is the title of the article associated with that user's first-ever edit on Wikipedia? The answer should be the article title, correctly capitalized.
**Analysis:**
Both tasks require digging through Wikipedia, but they emphasize different skills and sources: the original task hinges on interpreting the Legume page's public logs to identify the specific content violation and then mapping that violation to Wikipedia's "three core policies" as they were defined before December 2022, isolating what the "R" stands for; this involves policy literacy, time-bounded interpretation, and resolving potential ambiguity in log descriptions and policy acronyms. The evolved task is a chain of archival lookups: locate the original page move for "Legume," identify the user who performed it, navigate to that user's contribution history, and extract the title of the article from their first-ever edit with correct capitalization; this stresses precise provenance tracing, familiarity with page histories and user logs, and attention to naming conventions rather than policy interpretation.

Figure 13: Task evolution case 5.

**Case 6**

**Original Task:**
What writer is quoted by Merriam-Webster for the Word of the Day from June 27, 2022?
**Evolved Task:**
Visit the Merriam-Webster 'Word of the Day' page for June 27, 2022. According to the etymology provided in the 'Did You Know?' section, the word 'jingoism' originated in the context of a specific 19th-century war. Identify the full name of the primary treaty that officially and finally concluded this war. Your answer should be the name of the treaty.
**Analysis:**
Both tasks start from the same Merriam-Webster Word of the Day page for June 27, 2022, but they emphasize different skills: the original task is a direct fact lookup to identify which writer is quoted on that page—requiring accurate navigation and citation capture from a single source. The evolved task chains that page to external historical research: use the "Did You Know?" etymology to identify the specific 19th-century war linked to "jingoism," then determine which treaty officially and finally concluded that war, taking care to distinguish preliminary accords from the definitive settlement and to provide the treaty's full formal name. This adds steps of context extraction, cross-referencing, and precision in historical nomenclature beyond the initial page.

Figure 14: Task evolution case 6.

**Case 7**

**Original Task:**
Given this table defining * on the set S = a, b, c, d, e
|*|a|b|c|d|e|
|—|—|—|—|—|—|
|a|a|b|c|b|d|
|b|b|c|a|e|c|
|c|c|a|b|b|a|
|d|b|e|b|e|d|
|e|e|d|b|a|d|c|

provide the subset of S involved in any possible counter-examples that prove * is not commutative. Provide your answer as a comma separated list of the elements in the set in alphabetical order.

**Evolved Task:**
Consider a binary operation '*' defined on the set S = 0, 1, 2, 3, 4, 5, 6, 7. The operation is defined by the formula: x * y = (3*x + 5*y) mod 8. Determine the total number of ordered triplets (a, b, c), where a, b, and c are elements of S, for which the associative property fails, i.e., (a * b) * c is not equal to a * (b * c). Provide the final answer as a string containing a single integer.

**Analysis:**
Both tasks concern algebraic properties of binary operations but require different approaches: the original task is a table-based analysis on a five-element set to find any pair witnessing non-commutativity and then report the subset of elements involved—this is about reading a Cayley table, detecting asymmetries $xy \neq yx$, and aggregating the implicated symbols in sorted order. The evolved task shifts to an eight-element modular operation defined by $xy := (3x + 5y) \bmod 8$ and asks for a global count of associativity failures over all ordered triplets, which entails formulating and checking the associativity condition either by full enumeration of $8^3$ cases or by deriving algebraic criteria for when $(ab)c \neq a\,(b*c)$ holds; it emphasizes modular arithmetic, structural properties of linear-combination operations on $\mathbf{Z}_8$, and programmatic verification to ensure complete and accurate counting.

Figure 15: Task evolution case 7.

---

**Case 8**

**Original Task:**
You are a telecommunications engineer who wants to build cell phone towers on a stretch of road. In the reference file is a layout of the road and nearby houses. Each dash, "-", is a marker indicating a mile. Each capital H indicates a house located next to a mile marker, appearing above or below the stretch of road. Each cell phone tower can cover houses located next to the road within a 4-mile radius. Find the minimum number of cell phone towers needed to cover all houses next to the road. Your answer should be a positive numerical integer value.

**Evolved Task:**
You are a telecommunications engineer tasked with deploying cell towers on a 2D grid defined in the provided Excel file, 'deployment_grid.xlsx'. Your goal is to find the minimum total deployment cost to provide coverage to all houses.

Grid Rules: - The grid is defined in the 'DeploymentGrid' sheet of the Excel file. - Cell values represent the content of the grid: - 'H': A house that needs coverage. Towers cannot be built on these cells. - 'X': A no-build zone. Towers cannot be built on these cells. - 'G': Green terrain (Terrain Cost = 1) - 'Y': Yellow terrain (Terrain Cost = 3) - 'B': Blue terrain (Terrain Cost = 5)

Tower Types & Costs: - You have two types of towers available: 1. 'Pico Tower': Base Cost = 10, Coverage Radius = 2 2. 'Macro Tower': Base Cost = 30, Coverage Radius = 4 - The 'Total Deployment Cost' for a single tower is its 'Base Cost' + the 'Terrain Cost' of the cell it is placed on. - The total cost for the project is the sum of costs for all deployed towers.

Coverage Rules: - A tower at '(r1, c1)' covers a house at '(r2, c2)' if the Manhattan distance between them is less than or equal to the tower's radius. - Manhattan Distance = '|r1 - r2| + |c1 - c2|'.

Your task is to determine the absolute minimum total deployment cost to ensure every house on the grid is covered by at least one tower. The final answer should be a single integer representing this minimum cost.

**Analysis:**
The evolved task is substantially harder than the original: the original is a one-dimensional placement problem along a road with identical towers and a uniform 4-mile radius, reducible to a classic interval covering problem solvable in linear time by a simple greedy strategy that repeatedly places a tower as far right as possible within 4 miles of the leftmost uncovered house; by contrast, the evolved task operates on a two-dimensional Excel-defined grid with forbidden cells (H, X), buildable terrains (G/Y/B) that impart different terrain costs, and two tower types (Pico radius 2 with base cost 10, Macro radius 4 with base cost 30), where each tower's deployment cost is its base cost plus the terrain cost and coverage uses Manhattan distance, making the objective of minimizing total cost while covering all houses a weighted set cover/facility location problem: each buildable cell paired with a tower type is a candidate facility with a specific cost and coverage set, and we must select a minimum-cost subset covering every H, which is NP-hard in general and thus best modeled via an integer linear program with binary decision variables and coverage constraints or approximated with heuristics such as cost-effectiveness greedy or Lagrangian methods, all preceded by parsing the spreadsheet and precomputing coverage sets.

Figure 16: Task evolution case 8.

**Case 9**

**Original Task:**
Jen enters a lottery by picking $4$ distinct numbers from $S = \{1, 2, 3, \cdots, 9, 10\}$. $4$ numbers are randomly chosen from $S$. She wins a prize if at least two of her numbers were $2$ of the randomly chosen numbers, and wins the grand prize if all four of her numbers were the randomly chosen numbers. The probability of her winning the grand prize given that she won a prize is $\frac{m}{n}$ where $m$ and $n$ are relatively prime positive integers. Find $m + n$.

**Evolved Task:**
Jen picks a set $J$ of 4 distinct numbers from $S = \{1, 2, \ldots, 10\}$. Tom then picks a set $T$ of 4 distinct numbers from $S$, chosen uniformly at random from all such sets that have exactly one number in common with $J$ (i.e., $|J \cap T| = 1$). A lottery then randomly draws a set $L$ of 4 distinct numbers from $S$.

A 'J-only Prize' is awarded if the lottery set $L$ intersects Jen's set $J$ but is completely disjoint from Tom's set $T$. The probability of this prize being awarded can be expressed as a fraction $\frac{m}{n}$, where $m$ and $n$ are relatively prime positive integers. Find $m + n$.

**Analysis:**
The adapted problem is significantly more difficult and conceptually deeper than the original. The original problem is a standard exercise in combinations and conditional probability. The solution path is straightforward: one simply needs to calculate the number of outcomes for the winning event ("at least two matches") and for the grand prize event ("four matches"), then find their ratio. It primarily tests a student's fluency with combinatorial formulas. The adapted problem, however, introduces a higher level of complexity by adding a third set, T, with a specific relationship to J ($|J \cap T| = 1$). This transforms the problem from a simple counting exercise into one requiring sophisticated logical and set-theoretical reasoning. The key to solving it is not direct calculation, but the conceptual leap of partitioning the universal set S into four distinct subsets based on J and T. Once this structure is understood, the calculation becomes surprisingly simple. In essence, the difficulty shifts from mechanical computation to conceptual abstraction. The adapted problem is harder because it demands a deeper insight into the underlying structure of the sets involved, making it a more elegant and challenging mathematical puzzle.

Figure 17: Task evolution case 9.

---

**Case 10**

**Original Task:**

Alice and Bob play the following game. A stack of $n$ tokens lies before them. The players take turns with Alice going first. On each turn, the player removes either $1$ token or $4$ tokens from the stack. Whoever removes the last token wins. Find the number of positive integers $n$ less than or equal to 2024 for which there exists a strategy for Bob that guarantees that Bob will win the game regardless of Alice's play.

**Evolved Task:**

An impartial game is played with two stacks of tokens, with sizes $n$ and $m$. Two players take turns making moves. A move consists of removing $k$ tokens from *each* stack. The set of allowed values for $k$ depends on the current minimum stack size, $p = \min(n, m)$: - If $p < 20$, the allowed moves are $k \in \{1, 2, 4\}$. - If $p \geq 20$, the allowed moves are $k \in \{1, 4\}$.

A move is only possible if both stacks have at least $k$ tokens. The player who makes the last possible move wins. A starting position $(n, m)$ is a 'losing position' if the second player has a guaranteed winning strategy.

Find the total number of losing positions $(n, m)$ such that $1 \leq n \leq 100$ and $1 \leq m \leq 100$.

**Analysis:**

The leap in difficulty from the original to the adapted problem is enormous. The original is a standard one-dimensional game theory exercise where the solution hinges on discovering a simple, repeating pattern in the losing positions (P-positions). It primarily tests pattern recognition.

The adapted problem, while appearing to be a more complex two-dimensional game, is a sophisticated test of abstract reasoning. The critical insight, which is far from obvious, is that the game's two-dimensional state (n, m) collapses. The winning or losing status of any position is determined solely by the minimum of the two stacks, p = min(n, m), and which set of rules applies at that value of p.

This reduces the problem to a one-dimensional analysis again, but one that is masked by a misleading setup and complicated by conditional rules. The difficulty is therefore elevated from simple pattern-finding to a much higher level of model simplification and abstraction, followed by a significantly more complex counting phase. It's a shift from solving a problem to figuring out what the problem really is.

Figure 18: Task evolution case 10.

