# OpenReview forum: "Towards Self-Evolving Agent Benchmarks : Validatable Agent Trajectory via Test-Time Exploration"
_ICLR.cc/2026/Conference — ICLR 2026 Poster_

### Official Review · Reviewer_s49g · 2025-10-19

**Soundness:** 3
**Presentation:** 4
**Contribution:** 3
**Rating:** 8
**Confidence:** 5

**Summary:**

This paper proposes **TRACE**, an automated pipeline for generating new and harder tasks from existing benchmarks.  The system leverages a combination of LLM-based reasoning and validation to transform existing agent tasks into more complex variants, with the stated goal of alleviating benchmark saturation and stimulating progress in reasoning-heavy domains.  TRACE operates by (1) analyzing existing benchmarks such as GAIA, (2) proposing modifications that make the tasks more challenging, and (3) validating the modified tasks through an auxiliary solver before including them in the new dataset.  Empirically, the authors demonstrate that models perform worse on these generated tasks, suggesting that TRACE successfully creates more difficult evaluations.  The idea of using LLMs to automatically expand and stress-test benchmarks is promising and timely.

This is an exciting and well-motivated step toward automated benchmark evolution, but the methodology currently conflates task difficulty with model failure and lacks grounding in construct validity.   With better controls, clearer interpretability, and a modular release for community use, this work could become a genuinely valuable contribution to the field.

**Strengths:**

1. **Originality and relevance.**
   The notion of an *automatic benchmark generator* is highly relevant. Existing benchmarks saturate quickly, and manually curating new tasks is costly. TRACE offers a compelling automation framework that can rejuvenate established benchmarks by introducing more challenging variants.

2. **Methodological soundness (conceptually).**
   The pipeline’s modular design—comprising generation, validation, and filtering—provides a flexible structure that could be extended to multiple domains. In principle, this could evolve into a general-purpose benchmark-expansion tool.

3. **Potential for community impact.**
   If released as an open-source package, TRACE could become a widely adopted toolkit. It could serve as a plug-and-play module attached to popular agent benchmarks (e.g., GAIA, SWE-Bench, Terminal-Bench, WebArena).  I encourage the authors to release TRACE as a standalone, modular Python package or API so that the community can easily apply it to existing datasets.  Since some teams (e.g., Terminal Bench) have been building adapters to existing agent benchmarks, this is probably not too much work.  Doing so would make TRACE a genuine contribution of infrastructure and reproducibility.

4. **Empirical value.**
   Demonstrating that LLMs perform significantly worse on TRACE-generated tasks suggests that the system indeed increases difficulty and task complexity.  The paper shows that the generated tasks are longer, require more steps, and involve deeper reasoning than the originals.



Lastly, the command of English is excellent.  Good job.

**Weaknesses:**

Despite its creativity and potential, the methodology has a few central problems.  Problems 1 is the most severe since it undermines all data reported in the paper.  Problem 2 should also be fixed.  I don't mind if problem 3 is left unsolved.

---

### 1. Selection bias — selecting on what you are reporting

The main methodological flaw is **selection bias** in the task-filtering procedure.   The pipeline uses an “auxiliary validator” (the specific model is unclear to me—possibly Qwen3-Coder-480B?) and retains only tasks that this validator cannot solve.   Subsequently, the paper reports that LLMs perform worse on these filtered “new tasks” than on the original dataset.  This comparison is inherently **unfair** because the filtering criterion (unsolved by validator) directly ensures lower model performance.

To test whether the decreased performance is meaningful rather than tautological, I encourage the authors to perform a **control experiment**:  run the same filtering procedure on the original GAIA dataset, selecting only the tasks the validator cannot solve, and then compare model accuracy on this filtered subset.   I predict the same “performance drop” would appear even without TRACE-generated tasks, simply because of selection bias.

By the way, this problem mirrors the issue in *Humanity’s Last Exam (HLE)*: if one only includes tasks that all frontier models initially fail, one will trivially obtain a benchmark where all frontier models score poorly.  Such difficulty is not an intrinsic property of the tasks—it merely reflects transient weaknesses of current models.  As newer models emerge, the scores rise sharply, revealing that the selection mechanism was biased toward the temporary state of frontier capabilities.  This is what happened with HLE.  A few months later, when the newer models are released, there was an instead jump in scores.

To make the “decreased performance” claim meaningful, the authors should either:
- provide an **independent definition of difficulty**, not one derived from model failure; or
- report **orthogonal measures of benchmark quality**, such as separability with confidence (see the Agent-Hard paper) or correlation with human difficulty ratings.

---

### 2. Construct validity — realism and meaningfulness of new tasks

Construct validity is the core problem here. See the literature on measurement in general.  Goes back to Cronbach, L. J., & Meehl, P. E. (1955).  While the new tasks may indeed require more steps and longer trajectories, it is unclear whether they measure anything meaningful or realistic.  In benchmark design, *construct validity*—the degree to which a test measures what it purports to measure—is paramount.
A benchmark should not only resist saturation but also preserve *semantic relevance* and *economic relevant*.

Even thouhg SWE-Bench Verified has a lot of bugs, it has a clear external meaning: solving GitHub issues corresponds to resolving real software bugs.  If all models achieve $100%$ on GSM8K, we can still interpret that success as mastery of grade-school math reasoning.  If all models do well in telling dogs and cats apart, we can still understand the state of progress in Computer Vision.  But what exactly does it mean, conceptually or practically, for deepseek-v3 to achieve $24.7%$ on GAIA-TRACE versus $41.8%$ on GAIA?  If these scores cannot be linked to a real-world applicable capabilities, then the numbers risk becoming meaningless.

This issue parallels the design philosophy behind overly artificial exams (such as the Chinese Gao Kao): one can always construct harder and more complicated problems, but hardness alone does not confer value.  TRACE currently optimizes only for *difficulty* and *saturation resistance*, rather than for *meaningfulness* or *construct validity*.  Without grounding in a real-world construct, the benchmark may end up measuring the ability to chain API calls or manipulate evidence density—interesting but not inherently valuable.  How should you explain to my grandma, an outsider to the field, what this 24.7% mean here, if she wants to make an investment decision in AI agents?

To address this, I recommend that the authors:
- provide human evaluations of *task realism* and *relevance*;
- discuss construct validity of the benchmark in light of the measurement-theory literature;
- help readers understand why the new tasks are not just messier and more complicated to solve but also meaningful and realistic to expect an agent to solve.

This will encourage community adoption.

---

### 3. Missing comparisons and broader context

The paper would benefit from situating TRACE in relation to similar ideas.  For example, Jason Weston’s *Self-Challenge Agents* also propose generating new tasks automatically to train and evaluate agents.  How do TRACE’s generated tasks differ in structure, diversity, or usefulness?  Do these novel tasks help for training?  Benchmarks nowadays are not just used for measurements but also serve as RL training environments.  It'd be wonderful if TRACE can make the dataset of RL environments for agents larger, since we are all running out of high quality data nowadays.  Clarifying this comparison would strengthen the conceptual framing.

**Questions:**

1. **Auxiliary validator model.**
   What exact model serves as the validator? How sensitive are the results to its capacity (e.g., Qwen3-32B vs Qwen3-480B)?

2. **Independent validation of difficulty.**
   Could human annotators or external scoring metrics (e.g., model confidence, entropy, or step length) be used to verify that TRACE-generated tasks are indeed harder for legitimate reasons?

3. **Construct validity assessment.**
   How do the authors define the “meaningfulness” of a task?  Have they considered measuring correlations between TRACE scores and other established benchmarks of reasoning or problem-solving ability?

4. **Integration and reproducibility.**
   Will the authors release TRACE as an open-source toolkit that can plug into existing agent benchmarks (e.g., via a simple adaptor API)?  This would significantly enhance its impact and adoption.  I also hope that we can see if the new tasks can be turned into an RL environment.

---

> ### Author Response · Authors · 2025-11-21
>
> We thank you for your recognition of our work and for your valuable feedback. We take your comments very seriously and will respond to your questions one by one. We hope our responses can resolve your inquiries. We also look forward to this opportunity to exchange ideas and have further discussions on these topics.
>
> ### Weakness 1
> > "Selection bias — selecting on what you are reporting"
>
> Thank you for pointing out this issue. We acknowledge the concern regarding selection bias and have indeed given it some thought and discussion. We would like to explain our motivation in the following points:
>
> ### 1) About the definition of difficulty
> Our perspective is that 'difficulty' does not lend itself to a single, intuitive definition. Instead, we argue that it is a highly context-dependent concept, contingent on factors such as the problem's background, its intrinsic logical structure, and the specific capabilities of the solver. For instance, a mathematical conjecture may be stated more simply than a complex olympiad problem, yet its fundamental difficulty is entirely different. A problem might present a large volume of information but be solvable by a simple algorithm. Furthermore, the same task poses a different level of challenge to an agent equipped with powerful tools versus a more basic agent.
>
> In fact, the **Trajectory Validator** component in TRACE framework is explicitly designed to compare the solution processes of the original and evolved tasks. It analyzes their respective bottlenecks to provide a comparative assessment of difficulty and, critically, filters out any evolved tasks that fail to demonstrate an increase in complexity. This mechanism embodies the principle that difficulty is context-dependent and provides a relative measure of difficulty (though we acknowledge this method may also have some bias). The **Auxiliary Validator**, therefore, serves a complementary purpose: it provides a direct signal of absolute difficulty. We employ it as a final filtering stage to further elevate the benchmark's overall challenge. In practice, we choose the Qwen-235B-A22B-Instruct as the auxiliary model.
>
> To further address your concerns and demonstrate the generality of our approach, we have conducted an additional experiment by applying the TRACE framework to AIME-2024, a competition-level mathematics dataset. In this setup, we intentionally adapted the pipeline by **removing the auxiliary validator**, relying solely on the model's intrinsic reasoning and code generation capabilities. This experiment serves two purposes: first, it showcases that TRACE is a widely portable paradigm capable of evolving tasks across different domains (from open-world agents to pure mathematical reasoning); second, it verifies that our evolutionary mechanism remains effective even in a streamlined configuration without external environmental feedback. The details of this experiment are presented below.
>
> ### 2) About the lightweight experiment with AIME-2024
> To verify the effectiveness of our framework without an auxiliary model, we conducted an additional lightweight experiment by applying our framework to evolve AIME-2024. In this iteration, we removed the auxiliary model and relied solely on the agent's intrinsic knowledge to assess whether the generated task increased in difficulty compared to the original. While we acknowledge that this approach may still introduce some bias, we argue that such bias is inevitable whenever a specific model is used to drive task evolution.
>
> For this new experiment, we utilized the Gemini-2.5-pro model, with the cost per evolution round remaining below $25 USD. (Most of our new experiments are conducted with API).

---

> > ### Author Response · Authors · 2025-11-21
> >
> > **Model Evaluation Results on AIME-2024 benchmark**
> >
> > **ROUND 2**
> >
> > | Metric | DeepSeek-R1-Distill-Qwen-7B (Evo. ← Orig.) | DeepSeek-R1-Distill-Qwen-7B (Δ) | DeepSeek-R1-Distill-Qwen-32B (Evo. ← Orig.) | DeepSeek-R1-Distill-Qwen-32B (Δ) | Qwen3-235B-A22B (Evo. ← Orig.) | Qwen3-235B-A22B (Δ) |
> > | :--- | :--- | :--- | :--- | :--- | :--- | :--- |
> > | **Average Acc.** | **0.4933** ← 0.5667 | **(-0.0734)** | **0.6233** ← 0.7300 | **(-0.1067)** | **0.9033** ← 0.9400 | **(-0.0367)** |
> > | **Average Length**| 14853.4 ← 16890.5 | (-2037.1) | **10556.7** ← 9912.8 | **(+643.2)** | **21808.9** ← 19265.0 | **(+2543.9)** |
> >
> > **ROUND 4**
> >
> > | Metric | DeepSeek-R1-Distill-Qwen-7B (Evo. ← Orig.) | DeepSeek-R1-Distill-Qwen-7B (Δ) | DeepSeek-R1-Distill-Qwen-32B (Evo. ← Orig.) | DeepSeek-R1-Distill-Qwen-32B (Δ) | Qwen3-235B-A22B (Evo. ← Orig.) | Qwen3-235B-A22B (Δ) |
> > | :--- | :--- | :--- | :--- | :--- | :--- | :--- |
> > | **Average Acc.** | **0.3933** ← 0.5667 | **(-0.1734)** | **0.5333** ← 0.7300 | **(-0.1967)** | **0.7167** ← 0.9400 | **(-0.2233)** |
> > | **Average Length**| **21092.7** ← 16890.5 | **(+4202.2)** | **15140.4** ← 9912.8 | **(+5227.6)** | **27283.2** ← 19265.0 | **(+8018.2)** |
> >
> > We can observe that after four rounds of evolution, the model's performance on the benchmark has significantly declined, and the token count has noticeably increased. For instance, the Average Acc. of the Qwen3-235B-A22B model decreased by 22.33%, and the average token count increased by over 8000. These experimental data demonstrate the significant change in the difficulty of the benchmark questions.
> >
> > Although this experiment is lightweight, its findings are significant. Mathematical reasoning is widely recognized as a cornerstone of advanced AI capabilities, and performance on datasets like AIME serves as a critical indicator of an agent's logical and deductive power. Therefore, demonstrating TRACE's effectiveness in this domain provides meaningful evidence of its value.
> >
> > ### 3) About orthogonal measures of benchmark quality
> > We also adopted the number of tokens generated (note that we do not count the input tokens) by the agent during problem-solving as a proxy for difficulty. While we acknowledge that this is an intuitive rather than a strictly rigorous metric, we use it here to demonstrate the increase in difficulty exhibited by the new benchmark.
> >
> > The results are shown below:
> >
> > | Metric | GPT-5-Mini (Evo. ← Orig.) | GPT-5-Mini (Δ) | KIMI-K2 (Evo. ← Orig.) | KIMI-K2 (Δ) |
> > | :--- | :---: | :---: | :---: | :---: |
> > | **Avg. Length (Round 1)** | **4898.2** ← 2864.5 | **(+2033.7)** | **6609.0** ← 3389.6 | **(+3219.4)** |
> > | **Avg. Length (Round 2)** | **6275.7** ← 2864.5 | **(+3411.2)** | **8454.7** ← 3389.6 | **(+5065.1)** |
> >
> > As shown in the table, the number of output tokens generated by the agent to solve the new problems has significantly increased. We have already detailed the changes in token count for the new experiment on AIME-2024 in the preceding section.

---

> > > ### Author Response · Authors · 2025-11-21
> > >
> > > ### Weakness 2
> > > > " Construct validity — realism and meaningfulness of new tasks"
> > >
> > > We thank you for this highly insightful and thought-provoking question. We agree that the issue you raised is critical. Below, we address your concerns from the following perspectives and look forward to further discussion with you.
> > >
> > > ### 1) About the discussion of construct validity of new tasks
> > >
> > > Firstly, we believe that the construct validity of the new data generated by TRACE is closely tied to the reference benchmark. Specifically, we utilize the GAIA dataset, a benchmark designed for General AI Assistants. Its core objective is to evaluate the fundamental capabilities required for AI systems to address real-world problems, such as reasoning, multi-modal information processing, web browsing, and proficient tool usage. Consequently, the new tasks synthesized by TRACE inevitably retain the bottlenecks of the original tasks and extend them further. This reference-based task evolution minimizes the generation of meaningless tasks, demonstrating the preservation of semantic relevance.
> > >
> > > Secondly, although we did not explicitly discuss construct validity in the manuscript, we did deeply consider and debate such issues during the actual development of this work. A typical question we debated was whether longer search chains are genuinely more meaningful. On this matter, we concluded that the true bottleneck is not the length of the action chain, but rather 'The Burden of Discovery.' Under this premise, a problem where the entry point for action is difficult to locate holds more real-world significance than a problem with a long action chain but a clear solution path.
> > >
> > > To reflect this philosophy, we carefully examined the tasks in the GAIA dataset and proposed several capabilities that we believe a new benchmark must genuinely assess, such as:
> > >
> > > * A. Multiple-Source Conflict and Reconciliation (Breadth)
> > > * B. Long Evidence Chains with Structure (Depth)
> > > * C. Multi-Modal, Complex Media Comprehension
> > > * D. Domain Transfer to Specialized Contexts
> > > * E. Toolchain Planning and Dependency
> > >
> > > Demonstrations of these bottlenecks can be found in Figure 4 of the paper.
> > >
> > > Through this approach, we effectively retain the essence of the GAIA dataset in evaluating General AI capabilities, while incorporating specific extensions (e.g., prompting the agent to evolve tasks toward specialized domains), demonstrating the preservation of economic relevance.
> > >
> > > We will clarify to readers that if an agent effectively solves our synthesized benchmark, it demonstrates strong general capabilities and represents a breakthrough in handling the aforementioned task bottlenecks.
> > >
> > > ### 2) Human evaluations of task realism and relevance
> > >
> > > Due to the time constraints of the rebuttal period, we were unable to conduct a comprehensive human study. However, we have sought to address the core of this concern through existing quantitative and qualitative analyses in our paper, which we believe provide strong preliminary evidence that the evolved tasks are indeed more meaningful and realistic.
> > >
> > > #### 1. Our Case Studies Directly Illustrate Increased Realism and Relevance
> > >
> > > The case studies presented in the paper (e.g., Table 1, and Cases 1-8 in the Appendix) are our primary evidence for this claim. They show a consistent pattern: TRACE transforms relatively simple, often single-skill problems into complex, multi-step challenges that better mirror real-world problem-solving, often requiring an **integration of multiple distinct capabilities**.
> > >
> > > * **Increased Realism:** Many evolved tasks introduce realistic constraints absent in the original versions. A prime example is Case 8. The original task is a highly simplified, one-dimensional problem: finding the minimum number of identical cell towers needed to cover houses along a straight road. This is a classic interval covering problem solvable with a simple greedy algorithm.
> > >     TRACE transforms this into a complex, two-dimensional facility location problem that mirrors a genuine engineering and business scenario. The evolved task requires the agent to parse a 2D grid from an Excel file and contend with multiple, realistic constraints. The agent must make economic trade-offs by choosing between two different tower types ('Pico' and 'Macro') with varying base costs and coverage radii, and placing them on different types of terrain ('Green', 'Yellow', 'Blue') that incur different deployment costs. Furthermore, certain areas are designated as no-build zones. The objective is to find the absolute minimum total deployment cost to cover all houses, not just the number of towers. This shift from an abstract puzzle to a multi-variable cost-optimization problem is a clear example of how TRACE generates tasks that are far closer to real-world engineering challenges.

---

> > > > ### Author Response · Authors · 2025-11-21
> > > >
> > > > * **Increased Relevance:** The evolved tasks often measure more valuable and generalizable capabilities. The task transformation described in Table 1 (and detailed in Appendix Case 1) powerfully illustrates this. The original task is fundamentally a fact-retrieval exercise: an agent must locate a specific scientific paper and extract a single, pre-existing value—the volume of a fish bag mentioned in the text. This tests a useful but basic "librarian" skill.
> > > >     The evolved task elevates the challenge from information retrieval to applied problem-solving and design. The agent is no longer asked to find a number; it is tasked with designing the optimal fish bag. Using the formulas provided in the same paper, the agent must build a mathematical model for a cylinder's volume, subject to a material constraint (a fixed surface area). To succeed, the agent must perform calculus-based optimization to determine the ideal radius and height that maximize the volume. We argue that this leap—from a "librarian" skill to an "analyst" or "engineer" skill—is a direct increase in the relevance of the capability being measured. The ability to not just find information but to use it to model and solve an optimization problem is far more indicative of the practical, economic value we expect from advanced AI agents.
> > > >
> > > > #### 2. Quantitative Data Shows a Shift Towards More Integrative and Relevant Capabilities
> > > >
> > > > Our analysis of the capability distribution provides further, quantitative support for this argument. The "**Integrative Reasoning**" category represents complex tasks that do not rely on a single dominant skill but instead require a non-trivial synthesis and chaining of multiple capabilities (e.g., web search, file parsing, and coding) to reach a solution.
> > > >
> > > > **Table: Comparison of capability distribution ratios before and after TRACE evolution on GAIA.**
> > > >
> > > > | **Capability Category** | **GAIA Original (%)** | **Evolve Round 1 (%)** | **Capability Totals (%)** |
> > > > | :--- | :--- | :--- | :--- |
> > > > | Web browsing | 43.9 | 42.2 | 27.1 |
> > > > | Coding | 19.1 | 20.5 | 24.7 |
> > > > | Multi-modality | 17.1 | 4.8 | 8.2 |
> > > > | Diverse filetype reading | 16.0 | 20.5 | 28.2 |
> > > > | **Integrative Capabilities** | **4.0** | **12.0** | **11.8** |
> > > >
> > > > The data reveals two critical trends. First, there is a clear increase in tasks requiring deep-interaction skills like **Coding** and **Diverse Filetype Reading**. Second, and perhaps most significantly, the proportion of tasks demanding **Integrative Capabilities** **tripled from 4.0% to 12.0%** after just one evolution round.
> > > >
> > > > This dramatic increase shows that TRACE does not simply add more steps but systematically generates problems that demand a holistic and synthetic approach. These integrative tasks are a far better proxy for the complex, multifaceted challenges professionals face daily, which rarely fall into a single, neat capability bucket. This shift provides direct evidence that our benchmark is evolving to measure more **relevant**, realistic, and meaningful competencies.
> > > >
> > > > In summary, while we plan to conduct a formal human evaluation for the camera-ready version, we believe our existing qualitative and quantitative evidence already provides a strong case that TRACE enhances not just the difficulty, but also the realism and relevance of the benchmark tasks.

---

> > > > > ### Author Response · Authors · 2025-11-21
> > > > >
> > > > > ### Weakness 3
> > > > > > "Missing comparisons and broader context"
> > > > >
> > > > > Thank you for the suggestion to compare our work with similar studies. As you recommended, we will use the **Self-Challenge Agents** as a point of comparison to demonstrate the uniqueness of our TRACE framework and our particular understanding of this task.
> > > > >
> > > > > We believe the most significant shared concept between our TRACE framework and Self-Challenge Agents is that both aim for the agent to progressively replace the role of human experts in the process of benchmark construction or evolution. This point is explicitly mentioned in the introduction of the Self-Challenge Agents paper. Based on this idea, a further point of convergence is the shared objective of having the agent generate new task proposals based on its environmental exploration.
> > > > >
> > > > > In contrast, we highlight the following unique aspects of our TRACE framework:
> > > > >
> > > > > ### 1) From the perspective of the agentic environment
> > > > > Self-Challenge Agents utilizes two benchmarks: M3ToolEval and Tau-Bench. However, despite showing a degree of diversity, this environment remains **simulated in nature**. The tools the agent uses are relatively simple, and the feedback from the environment is clear and easy to process.
> > > > >
> > > > > In contrast, our **TRACE framework fully exposes the model to the real world**. The model can connect to the live internet through tools like web search and read multi-modal information using various other tools. Consequently, it finds all the resources to construct new tasks from the real world. This behavior closely mimics a human expert sitting in front of a computer to create data. It is for this very reason that we opted for the GAIA dataset, which is known for its comprehensiveness and real-world authenticity.
> > > > >
> > > > > ### 2) About the structure and diversity
> > > > > We argue that the TRACE framework produces tasks with significantly different structures and diversity compared to Self-Challenge Agents. Primarily, distinct environments lead to fundamental task differences. Furthermore, by using GAIA tasks as references, TRACE generates 'GAIA-style' outputs that include deep research, multimodal understanding, mathematical modeling, and puzzles. Thus, TRACE maintains a high consistency with the original GAIA dataset's structure and diversity.
> > > > >
> > > > > It is worth noting that TRACE can readily facilitate both **free exploration** and **targeted task generation** for agents in real-world environments.
> > > > > * **For free exploration:** Users can omit GAIA task references and provide optional directions, allowing the agent to synthesize new tasks freely.
> > > > > * **For targeted task generation:** Users can manually add examples and bottlenecks; for instance, one might specify: 'Create a task that evaluates the agent's web search and video understanding capabilities.'
> > > > >
> > > > > ### 3) About the usefulness
> > > > > We elaborate on the usefulness of the TRACE framework across several key dimensions.
> > > > >
> > > > > * **From the perspective of evaluation data:** TRACE not only provides new data (such as the agent tasks mentioned in our paper and supplementary experiments on competition-level math problems mentioned previously), but more importantly, facilitates the evolution and transfer of evaluation data. For instance, our case studies demonstrate that TRACE seamlessly integrates multiple bottlenecks—such as PDF reading, information retrieval, and mathematical modeling—to formulate new agent tasks. Furthermore, it enables benchmarks to cover a broader range of knowledge domains; for example, in our experiments with AIME, we observed that TRACE expanded the evaluation scope from high school curriculum to undergraduate-level calculus and algebra.
> > > > >
> > > > > * **From the perspective of training data:** TRACE provides a robust environment for **Reinforcement Learning (RL)**. The rationale is as follows: TRACE requires not only the generation of a task description but, crucially, the provision of a reasoning trajectory, tool invocations, code execution paths, and a final answer to solve the problem. Our validator agent performs a step-by-step verification of this path to ensure the final answer can be derived; tasks that fail this check are directly discarded. Consequently, once a new task is generated, TRACE inevitably provides a relatively reliable execution path and final answer as supervision signals for RL training. This directly embodies the **'Validate-by-Reproduce'** design philosophy featured in our paper's title.

---

> > > > ### Comment · Reviewer_s49g · 2025-11-24
> > > >
> > > > Regarding bottlenecks, you said figure 4 but I think you really meant figure 5.

---

> ### Author Response · Authors · 2025-11-21
>
> ### Regarding the question of open-sourcing, our answer is affirmative.
>
> **Yes, we are committed to releasing TRACE as a modular open-source toolkit featuring an adaptor API to easily plug into existing benchmarks.**
>
> Regarding your suggestion on RL, we fully agree. Since TRACE generates multi-step trajectories with verifiable outcomes, the new tasks are naturally suitable for constructing RL environments. We are exploring providing a Gym-compatible interface in our future release to facilitate this.
>
> Currently, we are finalizing the release by addressing several key areas to ensure high quality:
>
> * **Cost Optimization:** Given that agent outputs can occasionally be unstable, we are implementing more granular retry and feedback mechanisms. This aims to optimize token consumption and significantly reduce deployment costs for users.
> * **Tool Refinement:** We are upgrading the agent's toolset to better align with the dynamic requirements of benchmark evolution.
> * **Best Practices for Memory & Instructions:** We recognize the need to improve memory exemplars and system instructions and are working on providing a set of optimized 'best practices' to guide users.
> * **API Integration:** As per your suggestion, we are strictly standardizing the API support to ensure it serves as a flexible adaptor for various baselines.
>
> We are committed to resolving these issues to provide a robust and user-friendly open-source version for the community.

---

> > ### Comment · Reviewer_s49g · 2025-11-24
> > **Appreciate your responses.  They are excellent and I am impressed.**
> >
> > Keep up the good work!  The community needs people like you.

---

> > > ### Author Response · Authors · 2025-11-27
> > >
> > > We sincerely appreciate your supportive remarks and the recognition of our efforts. Your encouragement means a great deal to us and reinforces our commitment to advancing this research direction. Feedback like yours motivates us to further refine TRACE and continue contributing to the community with stronger methodologies and clearer insights. Thank you again for your thoughtful comments and valuable suggestions.

---

### Official Review · Reviewer_dJpA · 2025-10-25

**Soundness:** 3
**Presentation:** 3
**Contribution:** 2
**Rating:** 2
**Confidence:** 4

**Summary:**

The paper proposes TRACE, a self-evolving benchmark framework generating harder tasks via 3 stages (proposal mining, exploration, validation) with validatable trajectories. It enhances complexity/diversity, outperforms static benchmarks on GAIA. Paradigm shift to dynamic, self-evolving benchmarks enabling sustainable, adaptive, and continual agent evaluation progress.

**Strengths:**

1. The framework is carefully engineered and empirically validated, featuring a well-defined three-stage pipeline supported by detailed algorithmic design.

2. The paper is clear, written making it easy to follow the motivation, methodology, and outcomes.

**Weaknesses:**

1. While the paper presents an interesting multi-agent system for benchmark evolution, its distinct contribution relative to existing “agentic” methods is not sharply delineated.

2. Experiments are mostly centered on GAIA, with limited exploration across other benchmarks or domains.

3. The multi-agent pipeline (Proposer–Executor–Validator) introduces substantial systemic and computational overhead, which may limit practical deployment. The paper does not discuss resource consumption, time efficiency.

**Questions:**

none.

---

> ### Author Response · Authors · 2025-11-21
>
> We sincerely thank you for your constructive comments and for finding our work interesting. We address the three points raised below individually and trust that our responses and additional clarifications will effectively resolve your concerns.
>
> ---
>
> ### **Weakness 1**
> > "While the paper presents an interesting multi-agent system for benchmark evolution, its distinct contribution relative to existing “agentic” methods is not sharply delineated."
>
> We thank you for this opportunity to clarify the distinct positioning of **TRACE** within the landscape of agentic benchmark evolution. While recent works also utilize agents for this goal, TRACE introduces a fundamental paradigm shift from surface-level perturbation to trajectory-based exploration.
>
> Below, we delineate our contributions by comparing TRACE with the two key works:
>
> > **[1] Benchmark Self-Evolving: A Multi-Agent Framework for Dynamic LLM Evaluation**
> > This work is relevant as it also employs a multi-agent system to automatically evolve datasets.
>
> > **[2] AlphaEvolve: A coding agent for scientific and algorithmic discovery**
> > Although AlphaEvolve is not designed for data synthesis, it embodies the principle of solving complex problems through progressive evolution.
>
> ---
>
> ## Compared with Benchmark Self-Evolving [1]
>
> The multi-agent system in the related work [1] is composed of three types of agents: an Instance Pre-filter, an Instance Creator, and an Instance Verifier. Compared to our proposer-executor-validator structure, there are several key differences:
>
> ### 1) From the perspective of objective
> Our framework places a greater emphasis on **difficulty escalation**, whereas their work focuses more on enhancing **benchmark robustness**.
>
> * **First,** their Instance Pre-filter screens out problems that the agent cannot solve beforehand, which inherently lowers the benchmark's baseline difficulty. In contrast, our approach provides the agent with solution steps for foundational problems and requires it to analyze the bottleneck, rather than simply discarding difficult instances.
> * **Second,** their evolution process prioritizes robustness through methods such as question substitution, context rewriting, and inverting key information. Our framework, on the other hand, provides the agent with more explicit difficulty criteria and prior knowledge to increase complexity, such as increasing the amount of information required for a solution, and shifting problems to more specialized domains, or even introduce problems that require complex abilities (Specific examples can be seen in Table 1 and Figures 8-15 of the paper, and we have provided a detailed explanation for the evolution of each example).
> * **Third,** we incorporate a dedicated **Difficulty Validation mechanism**, a critical component absent in [1]. This step is essential because TRACE operates on a test-time exploration paradigm. Unlike static rewriting, agentic exploration is inherently stochastic; a new trajectory might differ in content without necessarily requiring higher cognitive load. Our Validator acts as a necessary filter, explicitly measuring complexity metrics (e.g., reasoning depth, tool chain length) to distinguish "harder" tasks from merely "different" ones. This ensures the benchmark achieves a monotonic increase in difficulty, rather than random horizontal drift.
>
> ### 2) From the perspective of evolution mechanism
> Our framework is designed to allow the agent to autonomously evolve benchmarks by leveraging real-world materials, its own insights, and user-provided prior knowledge, whereas the system in related work [1] functions more as a rule-based evolution system.
>
> * **Related Work [1]:** Specifically, Work [1] relies on rigid, predefined operations like Context Paraphrasing and Polarity Reversing. While effective for textual robustness, this approach constrains the agent's role to a rule-follower, limiting its potential to simple text-only domains like GSM8K.
> * **TRACE:** In contrast, TRACE grounds the agent in a rich, real-world environment. Equipped with web access, multimodal perception, and file I/O tools, our agents do not merely rewrite text; they emulate the workflow of a human expert designer. They actively source new data and verify logic across modalities. This allows TRACE to evolve complex, general-purpose benchmarks (exemplified by our application to GAIA) that evaluate multi-dimensional capabilities far beyond the scope of static text manipulation.
>
> *Related works such as Benchmark Self-Evolving is highly forward-looking and relevant. We will incorporate more discussion in a subsequent version and take the opportunity to emphasize our unique contributions.*

---

> > ### Author Response · Authors · 2025-11-21
> >
> > ## Compared with AlphaEvolve [2]
> >
> > First and foremost, we wish to highlight a core insight we share with the powerful agent system, **AlphaEvolve**: both our systems rely on the agent's intrinsic capabilities rather than on rule-based evolution. AlphaEvolve depends on the model's own ability to generate ideas for iterative evolution. In contrast, while our system incorporates human knowledge as high-level guidance, this guidance only outlines a general direction without dictating specific scenarios. In essence, we believe both approaches are manifestations of how LLMs can tackle complex tasks in agentic scenarios through test-time scaling.
> >
> > Our key differences are as follows:
> >
> > ### 1) From the perspective of interaction scope
> > **AlphaEvolve** operates primarily within a closed "propose-and-code" loop, optimizing algorithms in a sandbox. In contrast, **TRACE** targets open-ended, real-world tasks. Our agents utilize a comprehensive toolset (web browsing, file I/O, multimodal perception) to interact with external environments. While AlphaEvolve discovers algorithms, TRACE discovers information and scenarios, bridging the gap between abstract reasoning and real-world application.
> >
> > ### 2) From the perspective of method design
> >
> > * **Dedicated Proposal Agent:** Unlike AlphaEvolve's integrated loop, TRACE employs a dedicated agent for proposal generation. This structural decoupling allows the system to prioritize divergent pre-exploration, fostering a wider variety of evolutionary directions. Crucially, this agent performs a bottleneck analysis on the seed task to identify its core difficulties. This diagnostic step ensures that subsequent modifications are targeted and effective, rather than random perturbations.
> >
> > * **Advanced Validator for Open-Ended Tasks:** Unlike scientific discovery tasks, which often have reliable methods to validate evolutionary outcomes, our work focuses on open-ended tasks where evaluation must rely on the agent's own judgment. Consequently, we have invested more effort in designing our validator to assess newly generated problems:
> >     * A foundational aspect of our design is that the framework not only generates a new problem but also outputs the corresponding agent trace from its solution process.
> >     * **For correctness verification**, our key insight is to decouple fine-grained factual verification (e.g., checking if the output of a tool call in the agent trace matches the claimed result) from holistic logical verification. This separation enhances the reliability of the validation process.
> >     * **For difficulty assessment**, we introduce several metrics, such as whether a problem increases the information required for a solution or elevates the complexity of multimodal understanding, to provide the agent with a concrete basis for its evaluation.
> >
> > While AlphaEvolve represents a highly sophisticated and powerful system for scientific discovery, TRACE offers specialized innovations necessary for the distinct challenges of dataset synthesis. We hope this discussion effectively highlights the unique contributions and design rationale of our work.

---

> ### Author Response · Authors · 2025-11-21
>
> ### Weakness 2
> > "Experiments are mostly centered on GAIA, with limited exploration across other benchmarks or domains."
>
> Thank you for pointing out this shortcoming. We would like to offer an explanation and, to address your concerns, have conducted supplementary experiments on AIME-2024, a competition-level mathematics dataset.
>
> Our brief explanation is as follows: We chose GAIA[3] because it is a dataset specifically designed to evaluate the capabilities of generalist agents. These capabilities span a wide range, including video and image comprehension, audio and speech understanding, in-depth research, and mathematical modeling. This scope directly covers the multiple domains you mentioned.
>
> > [3] GAIA: a benchmark for General AI Assistants
>
> To further address your concerns and simultaneously validate the capabilities of TRACE in other domains, **we have followed your instructions to conduct an additional experiment by applying it to evolve problems from AIME-2024, a widely-used mathematics dataset**.
>
> In this experiment, our agent system, powered by the Gemini-2.5-pro model, was guided by a specific set of rules. The agent was instructed to adapt problems based on their core insights and bottlenecks of the original math problem. The objective was to elevate their elegant mathematical difficulty, rather than introducing superficial modifications (e.g., rephrasing that does not alter the problem's mathematical essence) or merely increasing the computational complexity. We also provide two adapted examples as case studies.
>
> We performed four rounds of evolution. The newly generated problems were then used to test three models: DeepSeek-R1-Distill-Qwen-7B, DeepSeek-R1-Distill-Qwen-32B, and Qwen3-235B-A22B. The increase in difficulty was quantified by the changes in average accuracy over 10 evaluation runs and the average number of response tokens.
>
> ---
>
> ## Model Evaluation Results on AIME-2024 benchmark
>
> ### ROUND 2
>
> | Metric | DeepSeek-R1-Distill-Qwen-7B (Evo. ← Orig.)| DeepSeek-R1-Distill-Qwen-7B (Δ)| DeepSeek-R1-Distill-Qwen-32B (Evo. ← Orig.)| DeepSeek-R1-Distill-Qwen-32B (Δ)| Qwen3-235B-A22B (Evo. ← Orig.) | Qwen3-235B-A22B (Δ) |
> | :--- | :--- | :--- | :--- | :--- | :--- | :--- |
> | **Average Acc.** | **0.4933** ← 0.5667 | **(-0.0734)** | **0.6233** ← 0.7300 | **(-0.1067)** | **0.9033** ← 0.9400 | **(-0.0367)** |
> | **Average Length** | 14853.4 ← 16890.5 | (-2037.1) | **10556.7** ← 9912.8 | **(+643.2)** | **21808.9** ← 19265.0 | **(+2543.9)** |
>
>
> ### ROUND 4
>
> | Metric | DeepSeek-R1-Distill-Qwen-7B (Evo. ← Orig.)| DeepSeek-R1-Distill-Qwen-7B (Δ)| DeepSeek-R1-Distill-Qwen-32B (Evo. ← Orig.)| DeepSeek-R1-Distill-Qwen-32B (Δ)| Qwen3-235B-A22B (Evo. ← Orig.) | Qwen3-235B-A22B (Δ) |
> | :--- | :--- | :--- | :--- | :--- | :--- | :--- |
> | **Average Acc.** | **0.3933** ← 0.5667 | **(-0.1734)** | **0.5333** ← 0.7300 | **(-0.1967)** | **0.7167** ← 0.9400 | **(-0.2233)** |
> | **Average Length** | **21092.7** ← 16890.5 | **(+4202.2)** | **15140.4** ← 9912.8 | **(+5227.6)** | **27283.2** ← 19265.0 | **(+8018.2)** |
>
> ---
>
> We can observe that after four rounds of evolution, the model's performance on the benchmark has significantly declined, and the token count has noticeably increased. For instance, the Average Acc. of the Qwen3-235B-A22B model decreased by 22.33%, and the average token count increased by over 8000. **These experimental results demonstrate a significant increase in the difficulty level of the benchmark questions**.

---

> ### Author Response · Authors · 2025-11-21
>
> ## Task Comparison
>
> | Original Task | Evolved Task |
> | :--- | :--- |
> | **Case 1: Probability Problem** -- Jen enters a lottery by picking $4$ distinct numbers from $S=\{1,2,3,\dots,9,10\}.$ $4$ numbers are randomly chosen from $S.$ She wins a prize if at least two of her numbers were $2$ of the randomly chosen numbers, and wins the grand prize if all four of her numbers were the randomly chosen numbers. The probability of her winning the grand prize given that she won a prize is $\frac{m}{n}$ where $m$ and $n$ are relatively prime positive integers. Find $m+n$. | **Case 1: Probability Problem** -- Jen picks a set $J$ of 4 distinct numbers from $S = \{1, 2, \dots, 10\}$. Tom then picks a set $T$ of 4 distinct numbers from $S$, chosen uniformly at random from all such sets that have exactly one number in common with $J$ (i.e., $\|J \cap T\| = 1$). A lottery then randomly draws a set $L$ of 4 distinct numbers from $S$. A 'J-only Prize' is awarded if the lottery set $L$ intersects Jen's set $J$ but is completely disjoint from Tom's set $T$. The probability of this prize being awarded can be expressed as a fraction $\frac{m}{n}$, where $m$ and $n$ are relatively prime positive integers. Find $m + n$. |
> | **Case 2: Game Theory Problem** -- Alice and Bob play the following game. A stack of $n$ tokens lies before them. The players take turns with Alice going first. On each turn, the player removes either $1$ token or $4$ tokens from the stack. Whoever removes the last token wins. Find the number of positive integers $n$ less than or equal to $2024$ for which there exists a strategy for Bob that guarantees that Bob will win the game regardless of Alice's play. | **Case 2: Game Theory Problem** -- An impartial game is played with two stacks of tokens, with sizes $n$ and $m$. Two players take turns making moves. A move consists of removing $k$ tokens from *each* stack. The set of allowed values for $k$ depends on the current minimum stack size, $p = \min(n, m)$. Specifically: (1) If $p < 20$, the allowed moves are $k \in \{1, 2, 4\}$; (2) If $p \ge 20$, the allowed moves are $k \in \{1, 4\}$. A move is only possible if both stacks have at least $k$ tokens. The player who makes the last possible move wins. A starting position $(n, m)$ is a 'losing position' if the second player has a guaranteed winning strategy. Find the total number of losing positions $(n, m)$ such that $1 \le n \le 100$ and $1 \le m \le 100$. |
> ### Brief Explanation: Why the Evolved Tasks Are Harder
>
> The evolved tasks are significantly more difficult because they introduce **additional constraints, multi-dimensional state spaces, and dynamic rules**. These changes demand a deeper level of insight and prevent the application of straightforward, standard formulas.
>
> * **For Case 1 (Probability):** The problem shifts from a standard conditional probability calculation to a complex scenario involving three interacting sets ($J, T, L$). The generation of set $T$ is **conditional** on $J$ ($|J \cap T| = 1$), and the final winning condition is a compound statement (intersecting with $J$ **and** being disjoint from $T$). This complexity requires more sophisticated combinatorial reasoning, such as the principle of inclusion-exclusion or careful casework, rather than simple counting.
>
> * **For Case 2 (Game Theory):** The problem evolves from a **one-dimensional** game solvable with simple modular arithmetic (a classic subtraction game) to a **two-dimensional** state space ($(n, m)$). Crucially, the game's rules are **dynamic** and state-dependent, changing based on the minimum stack size. This shatters the simple periodic pattern of the original game and forces a more complex analysis of the 2D state space, likely requiring piecewise reasoning or dynamic programming to identify the losing positions.
>
> We hope that our explanation and further experiments have addressed your questions.

---

> ### Author Response · Authors · 2025-11-21
>
> ### Weakness 3
> > "The multi-agent pipeline (Proposer–Executor–Validator) introduces substantial systemic and computational overhead, which may limit practical deployment. The paper does not discuss resource consumption, time efficiency."
>
> We appreciate the reviewer raising the important issue of efficiency. We acknowledge that our multi-agent pipeline is compute-intensive, but we argue this is a necessary trade-off for generating high-quality, verifiable benchmarks. We address this from three perspectives: empirical cost, the necessity of the multi-agent design, and our mitigation strategies.
>
> #### 1. Empirical Cost Report
> * **AIME-2024 (Pure Reasoning):** Evolving the dataset (one round) took approximately **30 minutes** and cost under **$25 USD** (using Gemini-2.5-pro). This is highly cost-effective for creating competition-level math problems.
> * **GAIA (General Agent):** One round took approximately **5 hours** using open-source models. While we did not track precise token counts for these specific runs, the higher latency is expected and inherent to the task nature—agents must interact with real websites, wait for server responses, and process multimodal files (PDFs/Images). This Test-Time Scaling cost is the price of grounding benchmarks in reality rather than synthetic text.
>
> #### 2. Why Multi-Agent?
> The primary advantage of a multi-agent system is its ability to decouple complex tasks, which prevents the performance degradation often caused by excessively long contexts. In practice, we frequently observe that a single agent is prone to oversights when handling multiple sub-tasks simultaneously. For instance, when a single validator is required to check the answer format, the correctness of tool calls, the logical process, and even compare the difficulty level all at once, it is highly likely to miss one of these aspects. Therefore, employing multiple agents for distinct roles is a necessary measure to enhance performance and improve stability.
>
> #### 3. Mitigation & Engineering Optimizations
> We have implemented several strategies to curb costs without sacrificing quality:
> * **Rollback Mechanism:** Instead of discarding a failed agent output, we implement state rollback, saving significant compute.
> * **Hybrid Compute:** We offload routine checks (e.g., format verification) to lightweight models or deterministic Python functions, reserving the expensive model generation only for complex reasoning and proposal mining.
> * **Tool Optimization:** For I/O heavy tasks, we optimized tool granularity (e.g., adding a `read_page` function for PDFs to avoid loading full documents), significantly reducing context window consumption.

---

> > ### Author Response · Authors · 2025-11-27
> >
> > Dear reviewer,
> >
> > Thank you for your insightful feedback, which is very helpful in improving the quality of our work.
> >
> > We hope that the clarifications and additional experiments in the revised draft sufficiently address the concerns you raised. We remain fully committed to addressing any remaining questions or points you would like to clarify during the discussion phase. Please feel free to reach out to us if you have any questions or additional feedback. We welcome further discussion.
> >
> > Thank you once more!

---

### Official Review · Reviewer_aYop · 2025-10-27

**Soundness:** 3
**Presentation:** 3
**Contribution:** 3
**Rating:** 4
**Confidence:** 4

**Summary:**

This paper proposes a method for the self-evolution of benchmarks, aiming to address the issue of benchmark performance saturation. Through multi-agent collaboration, the method further extends the difficulty and diversity of questions in existing benchmarks.

The approach consists of three stages, each carried out by a different agent:
* The first agent proposes potential directions for question evolution.
* The second agent generates evolved questions according to the proposed directions.
* The third agent verifies the feasibility and validity of the evolved questions.

This framework provides an automated mechanism for question expansion within benchmarks, and experimental results show that the evolved questions are significantly more challenging.

**Strengths:**

* This framework provides an automated mechanism for question expansion within benchmarks. The expansion process is fully automated, requiring no additional human annotation.
* Evaluation results show that the evolved tasks are significantly more challenging for models to complete, while the validation stage ensures that all evolved tasks remain solvable and valid.

**Weaknesses:**

A benchmark is not designed merely to make models score lower; rather, its purpose is to ensure that the scores reflect meaningful aspects of model capability. Therefore, this work appears to lack some necessary analyses to demonstrate that the benchmark evolution preserves this interpretability of scores.

* This work lacks a detailed statistical analysis of the evolved tasks — for example, in the style of GAIA, which provides statistics on the types of skills or capabilities required to complete each task.

* In addition, it does not provide a systematic analysis of the observed score degradation after benchmark expansion, which would help to clarify which aspects of task difficulty contribute most to the model's performance drop.

**Questions:**

* Could you include quantitative analyses of model capabilities after benchmark evolution? For instance, comparing the distribution of capability dimensions between the original and evolved benchmarks, or analyzing differences in token lengths of correct responses, could help clarify what specific aspects of model performance have degraded. Such statistics would make it clearer what the observed performance drop actually indicates about model weaknesses.

* Moreover, how to interpret changes in the ranking order of models across the original and evolved benchmarks? If the relative ranking of models shifts significantly, does it suggest that the evolved benchmark measures distinct or more nuanced abilities that were underrepresented before?

* Finally, the authenticity and validity of the evolved questions should be further examined. It remains unclear how the framework ensures that increased difficulty does not lead to artificial or ill-posed questions that merely make tasks harder without providing meaningful evaluation signals.

---

> ### Author Response · Authors · 2025-11-21
>
> We sincerely thank the reviewer for their thoughtful assessment and for recognizing our work We strictly value your feedback regarding the need for deeper statistical analysis and interpretability of the evolved tasks. Below, we respond to your specific questions point-by-point.
>
> ### Weakness 1
> > 'This work lacks a detailed statistical analysis of the evolved tasks.'
>
> Thank you for raising this question about how to more precisely characterize the change in model capabilities after benchmark evolution. We would like to offer an explanation and, to address your concerns, have conducted supplementary analyses on GAIA and its TRACE-evolved rounds.
>
> To address this, we followed your suggestion and measured, for each model and for each round (original GAIA, TRACE Round-1, TRACE Round-2), the average number of output tokens per question under our Pass@1 evaluation protocol. This metric is computed over all responses and reflects how long the model needs to “think” (i.e., how many reasoning and tool-interaction steps it executes) before producing an answer.
>
> As shown in the table below, there is a significant increase in the tokens generated by models (gpt-5-mini and kimi-k2) as they progress from the Original tasks to TRACE Evolved Rounds 1 and 2. This indicates that the evolved tasks force models to engage in much longer chains of reasoning and tool use, confirming that TRACE increases the demand for long-horizon planning in deep search problems.
>
> ### Token Length Results on GAIA-TRACE Benchmark
>
> | Metric | gpt-5-mini (Evo. $\leftarrow$ Orig.) | gpt-5-mini ($\Delta$) | kimi-k2 (Evo. $\leftarrow$ Orig.) | kimi-k2 ($\Delta$) |
> | :--- | :--- | :--- | :--- | :--- |
> | **Avg. Length (Evolved 1)** | **4898.2** $\leftarrow$ 2864.5 | **(+2033.7)** | **6609.0** $\leftarrow$ 3389.6 | **(+3219.4)** |
> | **Avg. Length (Evolved 2)** | **6275.7** $\leftarrow$ 2864.5 | **(+3411.2)** | **8454.7** $\leftarrow$ 3389.6 | **(+5065.1)** |
>
> ---
>
> To verify that this pattern extends beyond one benchmark, we applied our framework to AIME-2024, a rigorous competition-level math benchmark. Adhering to the same trajectory-based principles, we utilized a Gemini-2.5-pro–driven agent to restructure each problem around its core conceptual bottlenecks. This ensures the evolution targets deeper logical reasoning rather than merely inflating superficial arithmetic complexity.
>
> We then evaluated three reasoning-focused models—DeepSeek-R1-Distill-Qwen-7B, DeepSeek-R1-Distill-Qwen-32B, and Qwen3-235B-A22B—across the original and evolved rounds, tracking both average accuracy and response token length over 10 runs.
>
> ### Model Evaluation Results on AIME-2024 benchmark
>
> #### ROUND 2
>
> | Metric | DeepSeek-R1-Distill-Qwen-7B (Evo. $\leftarrow$ Orig.) | DeepSeek-R1-Distill-Qwen-7B ($\Delta$) | DeepSeek-R1-Distill-Qwen-32B (Evo. $\leftarrow$ Orig.) | DeepSeek-R1-Distill-Qwen-32B ($\Delta$) | Qwen3-235B-A22B (Evo. $\leftarrow$ Orig.) | Qwen3-235B-A22B ($\Delta$) |
> | :--- | :--- | :--- | :--- | :--- | :--- | :--- |
> | **Average Acc.** | **0.4933** $\leftarrow$ 0.5667 | **(-0.0734)** | **0.6233** $\leftarrow$ 0.7300 | **(-0.1067)** | **0.9033** $\leftarrow$ 0.9400 | **(-0.0367)** |
> | **Average Length** | 14853.4 $\leftarrow$ 16890.5 | (-2037.1) | **10556.7** $\leftarrow$ 9912.8 | **(+643.9)** | **21808.9** $\leftarrow$ 19265.0 | **(+2543.9)** |
>
> #### ROUND 4
>
> | Metric | DeepSeek-R1-Distill-Qwen-7B (Evo. $\leftarrow$ Orig.) | DeepSeek-R1-Distill-Qwen-7B ($\Delta$) | DeepSeek-R1-Distill-Qwen-32B (Evo. $\leftarrow$ Orig.) | DeepSeek-R1-Distill-Qwen-32B ($\Delta$) | Qwen3-235B-A22B (Evo. $\leftarrow$ Orig.) | Qwen3-235B-A22B ($\Delta$) |
> | :--- | :--- | :--- | :--- | :--- | :--- | :--- |
> | **Average Acc.** | **0.3933** $\leftarrow$ 0.5667 | **(-0.1734)** | **0.5333** $\leftarrow$ 0.7300 | **(-0.1967)** | **0.7167** $\leftarrow$ 0.9400 | **(-0.2233)** |
> | **Average Length** | **21092.7** $\leftarrow$ 16890.5 | **(+4202.2)** | **15140.4** $\leftarrow$ 9912.8 | **(+5227.6)** | **27283.2** $\leftarrow$ 19265.0 | **(+8018.2)** |
>
> Across rounds, all three models exhibit a **substantial drop in accuracy** together with a **clear increase in average response length**; for instance, on Round-4, Qwen3-235B-A22B loses over 22 percentage points of accuracy while its average output grows by more than 8k tokens. This mirrors the pattern we observe on GAIA/TRACE: as tasks are evolved, models both perform worse and are compelled to produce much longer reasoning traces.
>
> These quantitative analyses clarify the nature of the performance degradation. The data shows that models are expending significantly more cognitive resources (evidenced by longer token generation) but failing to maintain the necessary reasoning consistency over these longer horizons. This quantitatively supports our claim that TRACE effectively exposes weaknesses in long-context planning and sustained reasoning robustness rather than merely introducing surface-level noise.

---

> ### Author Response · Authors · 2025-11-21
>
> ### Weakness 2
> > 'It does not provide a systematic analysis of the observed score degradation after benchmark expansion... regarding changes in model ranking across original vs. evolved benchmarks.'
>
> Thank you for this insightful question regarding the shifts in model rankings between the original and evolved benchmarks. We appreciate you highlighting this aspect, as understanding *why* rankings change is just as important as the performance drop itself. We have conducted a detailed analysis to attribute these shifts to specific changes in task nature and difficulty, and we would like to explain our findings below.
>
> Our view is that these changes are not arbitrary, but arise from **(i) systematically lengthened, tool-intensive reasoning trajectories** and **(ii) a re-weighting of capability demands across task categories**.
>
> To make this concrete, we first analyze token-level behavior across evolution rounds. For each model and each split (GAIA original, TRACE Round-1, TRACE Round-2), we compute the average number of output tokens per question under our pass@1 protocol. With decoding hyperparameters fixed, we observe a consistent increase in token length in test-time computing as the benchmark evolves (as shown in the **Token Length Results on GAIA-TRACE Benchmark** table provided in the previous section). Models must “stay in the loop” for more steps, producing longer chains of reasoning, more tool invocations, and more self-corrections before reaching a final answer. This is precisely the signature of long-horizon, tool-heavy trajectories that are only weakly exercised by shorter, single-tool questions.
>
> Complementing this, we examine how the capability distribution of tasks shifts before and after evolution. Each instance is labeled by its dominant capability category (web browsing, coding, multi-modality, diverse filetype reading, Integrative Capabilities), and we report the ratios for TRACE Round-1/Round-2 in the table below (and in the per-model radar plots in the appendix).
>
> **Table: Comparison of Capability Distribution Ratios**
>
> | Capability Category | GAIA Original (Ratio) | Evolve Round 1 (Ratio) | Capability Totals (Ratio) |
> | :--- | :--- | :--- | :--- |
> | **Web browsing** | 43.9% | 42.2% | 27.1% |
> | **Coding** | 19.1% | 20.5% | 24.7% |
> | **Multi-modality** | 17.1% | 4.8% | 8.2% |
> | **Diverse filetype reading** | 16.0% | 20.5% | 28.2% |
> | **Integrative Capabilities** | 4.0% | 12.0% | 11.8% |
>
> As TRACE evolves GAIA, **Coding** and **Diverse Filetype Reading** tasks become more prominent, while pure web-browsing and lightweight multimodal lookup tasks are relatively down-weighted. Different models exhibit different capability features: some excel on web and simple multimodal tasks, others are comparatively stronger on coding-centric and structured document-processing tasks. When the benchmark re-weights toward the latter categories and demands longer trajectories, models whose strengths align with these more complex capabilities retain or improve their standing, while others drop.
>
> Taken together, the token-length and capability-distribution analyses support exactly the interpretation suggested by the reviewer: the evolved benchmark is measuring more nuanced, previously underrepresented abilities—such as long-horizon, tool-heavy reasoning and structured document processing—and thereby reveals qualitative differences between models that were partially obscured on the original GAIA.

---

> ### Author Response · Authors · 2025-11-21
>
> ### Response to Question on Authenticity and Validity of Evolved Questions
>
> > 'The authenticity and validity of the evolved questions should be further examined. It remains unclear how the framework ensures that increased difficulty does not lead to artificial or ill-posed questions...'
>
> We fully agree with the reviewer that increased difficulty is only meaningful if the resulting tasks remain authentic, well-posed, and provide reliable evaluation signals. TRACE explicitly incorporates this requirement at both the problem-formation and validation stages.
>
> First, our framework operates on **transparent, executable trajectories**: for every instance, we record a full sequence of (thought, action, observation) tuples, exposing each model decision and tool call rather than treating the task as a black-box prompt. The evolved question is derived from this trajectory and is therefore grounded in a concrete chain of reasoning and interactions, not in an arbitrary “make it harder” natural-language edit.
>
> Second, **Stage 2 (Question Formulation)** is explicitly constrained by three principles: authenticity, logical integrity, and solvability. The evolved problem must obey strict principles of authenticity, logical integrity, and solvability as encoded in the system prompt and algorithmic design (Fig. 5). Concretely, when a trajectory is converted into a new task, the formulation agent must:
> 1.  Stay grounded in the same real-world source(s) as the original GAIA instance (e.g., the same paper, document, or website).
> 2.  Ensure that the final answer is single-valued and deterministically determined by those sources.
> 3.  Avoid introducing free parameters or hidden assumptions that would make the problem ambiguous or under-specified.
>
> Tasks that violate these constraints (e.g., multiple plausible answers, dependence on stylistic choices rather than factual content) are rejected.
>
> Third, **Stage 3 (Multi-level Validation)** further enforces validity through three complementary checks:
> 1.  **Static Logical Check:** A review of the entire solution trace.
> 2.  **Dynamic Re-execution:** Running all code and tool calls to ensure observations are reproducible from scratch.
> 3.  **Integrity and Difficulty Audit:** Verifying the problem has a well-defined answer and represents a genuine increase in complexity over the original seed.
>
> Any candidate that is logically inconsistent, non-reproducible, or answer-ambiguous is discarded and never enters the final benchmark. As reported in the appendix, only a minority of model-generated candidates survive all three layers of filtering, underscoring that this validator is highly selective rather than permissive. Overall, TRACE is designed as a **validated-by-reproduce** framework: every accepted question is anchored to a reproducible trajectory, in contrast to benchmarks that rely solely on prompt-generated, superficially more complicated question text.
>
> ---
>
> ### Case Study: Evolved Tasks Necessitate Long-Horizon Planning and Dynamic System Modeling
>
> The example of the Eliud Kipchoge task (shown in Case 1 in the Appendix) directly addresses the concern regarding artificiality. The evolution does not merely "complicate" the question; it fundamentally shifts the evaluation paradigm from simple arithmetic to **long-horizon planning within a dynamic system**.
>
> **Testing Long-Horizon Planning and Multi-Step Reasoning:**
> TRACE transforms this into a **dynamic physical system** requiring the agent to first retrieve and compare Eliud Kipchoge’s 2018 and 2022 records to establish baselines. Then, by introducing the Moon’s recession, the agent must model **coupled variables**—where the target distance shifts over time—ultimately requiring **calculus-based analysis** to integrate these differential changes. This shift in complexity mandates a significant increase in **planning capabilities**. As shown in the solution trace (Steps 1-7), the agent cannot solve this in a single pass. It requires **long-horizon planning** to decompose the problem into a coherent dependency chain:
>
> 1.  Retrieve and normalize distinct historical data points (2018 vs. 2022 records).
> 2.  Compute baseline travel times for both scenarios.
> 3.  Calculate the secondary effect (recession distance) based on those baselines.
> 4.  Integrate these differential values into a final integer result.
>
> This structure forces the agent to engage in **more rounds of internal dialogue and planning**, maintaining the context of intermediate variables (like the specific time difference) over a longer solution path. A model that lacks long-horizon attention or logical rigor will fail to integrate the recession factor or confuse the two paces. Thus, the increased difficulty provides a distinct, meaningful signal regarding the agent's capacity for complex, multi-stage problem solving.
>
> We hope that our explanation and further experiments have addressed your questions.

---

> ### Comment · Reviewer_aYop · 2025-11-24
> **I'll raise my score to 6**
>
> Thanks for the detailed response.
>
> There are a few open-ended questions I’d like to discuss further. It’s noticeable that the capability-requirement distribution shifts significantly after expansion. Since GAIA is mainly sampled from real scenarios, we can reasonably view it as an effective approximation of real-world distributions. In your opinion, is it important to maintain the distribution as stable as possible during expansion? Or would such distributional shifts undermine the usefulness or interpretability of the resulting ranking scores?
>
> In any case, you’ve largely resolved my concerns, and I believe this paper deserves a higher score given the detailed clarifications you provided. I’ll raise my score to 6. Wish you the best of luck!

---

> > ### Author Response · Authors · 2025-11-27
> >
> > **Response regarding Data Distribution Maintenance:**
> >
> > We are grateful for your recognition of our work and for the valuable insights regarding
> > data distribution. We have given this significant thought and would like to share our
> > perspective on balancing "distributional fidelity" with "realistic evolution."
> >
> > **1. Preserving Fundamental Capability Requirements:**
> > We agree that maintaining a core distribution of capabilities similar to GAIA is important.
> > GAIA’s original tasks represent fundamental real-world skills expected of agents—such as
> > extracting information from PDFs/PPTXs, precise web searching, and writing code to solve
> > problems. These form the cornerstone for handling more complex tasks.
> >
> > Fortunately, this distribution can be actively managed within our framework. TRACE can be
> > directed via prompting to evolve tasks along specific capability dimensions. Furthermore,
> > the validation agent serves as a gatekeeper, verifying whether a new task effectively
> > assesses the intended capability, thereby filtering out deviations that do not meet these
> > fundamental requirements.
> >
> > **2. Strict Distributional Replication is Not Essential:**
> > While core capabilities should be preserved, we argue that **mechanically replicating** the exact
> > statistical distribution of the original GAIA dataset is neither necessary nor optimal, for two reasons:
> >
> > * **Relevance over Representation:** Some GAIA tasks (e.g., logic puzzles or ciphertext decoding),
> >     while interesting, represent niche skills rather than essential daily applications. It is sufficient
> >     to include a representative sample of these without strictly adhering to their original proportion.
> >     Instead, we prioritize augmenting tasks that better reflect high-value human needs.
> >
> > * **Bridging the "Real-World Gap":** There is a distinction between the current distribution of
> >     benchmark tasks and the complexity of real-world demands. Real-world tasks are often significantly
> >     more complex (e.g., a multi-constraint train scheduling problem vs. a simple coding exercise)
> >     and open-ended. Strictly adhering to GAIA's original distribution might limit our ability to
> >     model these higher-order complexities.
> >
> > **Conclusion: Purposeful and Calibrated Evolution**
> > In summary, when expanding a benchmark, the goal should not be to blindly mirror the original data
> > distribution, but to pursue a **purposeful and calibrated evolution**. We aim to retain the
> > fundamental capabilities GAIA represents (ensured by TRACE's design) while allowing for
> > **controlled distributional shifts**.
> >
> > We believe that such shifts do not undermine the validity of the ranking scores; rather, they
> > **enhance** the benchmark's guiding value by bridging the gap between existing datasets and the
> > complex, multifaceted demands of practical application. Thank you again for this insightful question;
> > it touches on the core philosophy of how next-generation benchmarks should be constructed.

---

### Official Review · Reviewer_H9KX · 2025-11-01

[review text omitted: it was posted to a different submission]

---

> ### Author Response · Authors · 2025-11-21
>
> We sincerely thank the reviewer for the positive evaluation and for promptly updating the review comments. Please don't worry about the mix-up at all. We fully understand how demanding the review process can be. In fact, we had initially made a diligent effort to interpret and address the previous comments within the context of our TRACE framework. However, we are now pleased to turn our full attention to your updated, specific feedback.
>
> Below, we provide a comprehensive, point-by-point response to these new questions. We hope that our response will resolve your concerns.
> ### Weakness 1
> > “The authors does not analyze why model performance drops on the evolved tasks. Does the decline truly reflects increased difficulty? Or simply stems from the synthesized tasks drifting away from realistic scenarios.”
>
> Thank you for your insightful question. We will now respond respectively to your two areas of concern. The first pertains to whether the degradation in agent performance demonstrates a genuine increase in the difficulty of the synthesis problem. The second concerns the validity and realism of the synthesis task, which can also make the performance drop.
>
> ### 1) About the definition and metric of 'difficulty'
>
> Firstly, we would like to talk about the definition of 'difficulty'. Our perspective is that 'difficulty' does not lend itself to a single, intuitive definition. Instead, we argue that it is a highly context-dependent concept, contingent on factors such as the problem's background, its intrinsic logical structure, and the specific capabilities of the solver.
>
> * For instance, a mathematical conjecture may be stated more simply than a complex olympiad problem, yet its fundamental difficulty is entirely different.
> * A problem might present a large volume of information but be solvable by a simple algorithm.
> * Furthermore, the same task poses a different level of challenge to an agent equipped with powerful tools versus a more basic agent.
>
> Therefore, in our view, difficulty cannot be measured with a precise numerical value, and the performance decline is an intuitive, proxy indicator.

---

> > ### Author Response · Authors · 2025-11-21
> >
> > ### 2) Supplementary metric and experiments demonstrating increased difficulty
> >
> > To enhance the credibility of the increased difficulty, we analyzed another intuitive proxy metric: the number of tokens the agent generated to solve new tasks, which serves to demonstrate the change in task difficulty.
> >
> > We even supplemented this with an additional experiment: using the TRACE framework to evolve AIME-2024, a popular mathematics benchmark, and then measuring the decline in accuracy across different reasoning models and the change in the number of tokens used for inference.
> >
> > **Token Count Statistics for the GAIA-Evolved Dataset:**
> >
> > | Metric | GPT-5-Mini<br>(Evo. $\\leftarrow$ Orig.) | GPT-5-Mini<br>($\\Delta$) | KIMI-K2<br>(Evo. $\\leftarrow$ Orig.) | KIMI-K2<br>($\\Delta$) |
> > | :--- | :---: | :---: | :---: | :---: |
> > | **Avg. Length (Round 1)** | **4898.2** $\\leftarrow$ 2864.5 | **(+2033.7)** | **6609.0** $\\leftarrow$ 3389.6 | **(+3219.4)** |
> > | **Avg. Length (Round 2)** | **6275.7** $\\leftarrow$ 2864.5 | **(+3411.2)** | **8454.7** $\\leftarrow$ 3389.6 | **(+5065.1)** |
> >
> > As shown in the table, the number of output tokens generated by the agent to solve the new problems has significantly increased.
> >
> > **Token Count and Accuracy Statistics for the AIME-Evolved Dataset:**
> >
> > **ROUND 2**
> >
> > | Metric | DeepSeek-R1-Distill-Qwen-7B<br>(Evo. $\\leftarrow$ Orig.) | DeepSeek-R1-Distill-Qwen-7B<br>($\\Delta$) | DeepSeek-R1-Distill-Qwen-32B<br>(Evo. $\\leftarrow$ Orig.) | DeepSeek-R1-Distill-Qwen-32B<br>($\\Delta$) | Qwen3-235B-A22B<br>(Evo. $\\leftarrow$ Orig.) | Qwen3-235B-A22B<br>($\\Delta$) |
> > | :--- | :--- | :--- | :--- | :--- | :--- | :--- |
> > | **Average Acc.** | **0.4933** $\\leftarrow$ 0.5667 | **(-0.0734)** | **0.6233** $\\leftarrow$ 0.7300 | **(-0.1067)** | **0.9033** $\\leftarrow$ 0.9400 | **(-0.0367)** |
> > | **Average Length** | 14853.4 $\\leftarrow$ 16890.5 | (-2037.1) | **10556.7** $\\leftarrow$ 9912.8 | **(+643.2)** | **21808.9** $\\leftarrow$ 19265.0 | **(+2543.9)** |
> >
> > **ROUND 4**
> >
> > | Metric | DeepSeek-R1-Distill-Qwen-7B<br>(Evo. $\\leftarrow$ Orig.) | DeepSeek-R1-Distill-Qwen-7B<br>($\\Delta$) | DeepSeek-R1-Distill-Qwen-32B<br>(Evo. $\\leftarrow$ Orig.) | DeepSeek-R1-Distill-Qwen-32B<br>($\\Delta$) | Qwen3-235B-A22B<br>(Evo. $\\leftarrow$ Orig.) | Qwen3-235B-A22B<br>($\\Delta$) |
> > | :--- | :--- | :--- | :--- | :--- | :--- | :--- |
> > | **Average Acc.** | **0.3933** $\\leftarrow$ 0.5667 | **(-0.1734)** | **0.5333** $\\leftarrow$ 0.7300 | **(-0.1967)** | **0.7167** $\\leftarrow$ 0.9400 | **(-0.2233)** |
> > | **Average Length** | **21092.7** $\\leftarrow$ 16890.5 | **(+4202.2)** | **15140.4** $\\leftarrow$ 9912.8 | **(+5227.6)** | **27283.2** $\\leftarrow$ 19265.0 | **(+8018.2)** |
> >
> > We can observe that after four rounds of evolution, the model's performance on the benchmark has significantly declined, and the token count has noticeably increased. For instance, the Average Acc. of the Qwen3-235B-A22B model decreased by 22.33%, and the average token count increased by over 8000. These experimental data demonstrate the significant change in the difficulty of the benchmark questions.

---

> > > ### Author Response · Authors · 2025-11-21
> > >
> > > ### 3) Qualitative Analysis of Realism for the Synthesis Task
> > >
> > > Due to the time constraints of the rebuttal period, we were unable to conduct a comprehensive human study. However, we have sought to address the core of this concern through existing quantitative and qualitative analyses in our paper, which we believe provide strong preliminary evidence that the evolved tasks are indeed more meaningful and realistic.
> > >
> > > The case studies presented in the paper (e.g., Table 1, and Cases 1-8 in the Appendix) are our primary evidence for this claim. They show a consistent pattern: TRACE transforms relatively simple, often single-skill problems into complex, multi-step challenges that better mirror real-world problem-solving, often requiring an **integration of multiple distinct capabilities**.
> > >
> > > * **Many evolved tasks introduce realistic constraints absent in the original versions.** A prime example is **Case 8**. The original task is a highly simplified, one-dimensional problem: finding the minimum number of identical cell towers needed to cover houses along a straight road. This is a classic interval covering problem solvable with a simple greedy algorithm. TRACE transforms this into a complex, two-dimensional facility location problem that mirrors a genuine engineering and business scenario.  The evolved task requires the agent to parse a 2D grid from an Excel file and contend with multiple, realistic constraints. The agent must make economic trade-offs by choosing between two different tower types ('Pico' and 'Macro') with varying base costs and coverage radii, and placing them on different types of terrain ('Green', 'Yellow', 'Blue') that incur different deployment costs. Furthermore, certain areas are designated as no-build zones. The objective is to find the absolute minimum total deployment cost to cover all houses, not just the number of towers. This shift from an abstract puzzle to a multi-variable cost-optimization problem is a clear example of how TRACE generates tasks that are far closer to real-world engineering challenges.
> > >
> > > * **The evolved tasks often measure more valuable and generalizable capabilities.** The task transformation described in Table 1 (and detailed in Appendix Case 1) powerfully illustrates this. The original task is fundamentally a fact-retrieval exercise: an agent must locate a specific scientific paper and extract a single, pre-existing value—the volume of a fish bag mentioned in the text. This tests a useful but basic "librarian" skill. The evolved task elevates the challenge from information retrieval to applied problem-solving and design. The agent is no longer asked to find a number; it is tasked with designing the optimal fish bag.  Using the formulas provided in the same paper, the agent must build a mathematical model for a cylinder's volume, subject to a material constraint (a fixed surface area). To succeed, the agent must perform calculus-based optimization to determine the ideal radius and height that maximize the volume. We argue that this leap—from a "librarian" skill to an "analyst" or "engineer" skill—is a direct increase in the relevance of the capability being measured. The ability to not just find information but to use it to model and solve an optimization problem is far more indicative of the practical, economic value we expect from advanced AI agents.
> > >
> > > ### 4) The TRACE Framework's Approach to Handling Difficulty and Realism in New Tasks
> > >
> > > We incorporate a dedicated **Difficulty Validation mechanism** in TRACE. Agentic exploration is inherently stochastic and a new trajectory might differ in content without necessarily requiring higher difficulty. Our Validator acts as a necessary filter, explicitly measuring complexity metrics (e.g., reasoning depth, tool chain length) to distinguish "harder" tasks from merely "different" ones. This ensures the benchmark achieves a monotonic increase in difficulty, rather than random horizontal drift.
> > >
> > > Furthermore, we try to ensure the realism of the evolved tasks through a dual-pronged approach:
> > > 1.  **Inherited Realism:** Our framework initializes evolution based on established, high-quality benchmarks like GAIA. By seeding the process with scenarios already validated for their real-world relevance, we ensure that every evolved task inherits a coherent, practical semantic foundation rather than starting from synthetic noise.
> > > 2.  **Real-World Grounding:** We constrain the evolution process to be material-driven. The TRACE framework tries to derive any new problem conditions or complexities from verifiable, real-world materials (e.g., actual web content, documents) encountered during the agent's exploration. This requirement also tries to prevent the system from hallucinating arbitrary difficulties, ensuring that the evolved tasks reflect the authentic complexity of the real world.
> > >
> > > We hope the explanations on the four points above have addressed your questions.

---

> ### Author Response · Authors · 2025-11-21
>
> ### Weakness 2 and Question
> > Regarding the solvability of the synthesized tasks, it would benefit from additional empirical evidence to support the claim. - The authors state that evolved tasks are solvable, but I was confused how this is guaranteed? For example, if the original task uses website A but the evolved task requires non-exist website B, how do you detect and filter out such unsolvable cases?
>
> This is a very profound and insightful question. We believe this question and Weakness 2, which you pointed out, are both related to the solvability of new tasks. Therefore, we will address them together here.
>
> We have also given thorough consideration to the solvability of new tasks and have implemented special mechanisms to address this issue. The solution we propose is the **Validate-by-Reproduce Paradigm** and the **Multi-Level Validation mechanism**.
>
> ### 1) Validate-by-Reproduce Paradigm:
>
> We propose this paradigm based on the following logic: if the reasoning behind each action in the generated agent trajectory is correct, and the actual outcome of each tool call or code execution matches its claimed result, then the agent task is, to a great extent, solvable, and its claimed result can be considered the final answer.
>
>
>
> Therefore, we abandoned the approach of having a validator directly verify the entire agent trajectory. Instead, we provide only a single step at a time—including the reasoning process, the tool call, and its claimed result—allowing the validator to check the reasoning logic and reproduce the tool call to see if it yields the same outcome. This method also avoids the greater risk of errors associated with validating a complete trajectory at once.
>
> Furthermore, after these step-by-step checks are completed, we have the validator examine the entire logical chain for overall coherence. Since the outcome of each step is already considered correct at this point, this final review serves to guarantee the task's solvability.
>
> ### 2) Multi-Level Validation Mechanism:
>
> Our validation system is multi-layered. It includes not only large model-based validators, but also lightweight validators built on Python scripts or smaller models.
>
>
>
> These lightweight validators handle checks such as: whether the generated content conforms to a predefined JSON format, if a claimed URL in a new task is valid, and whether a generated local file exists and its contents meet the specified requirements. By deploying these lightweight validators, we reduce the main agent's workload and significantly enhance the system's overall robustness.
>
> The issue you mentioned regarding a non-existent website can be ruled out by these lightweight validators. Moreover, even during the trajectory inspection process, such an error would be filtered out because the claimed result (e.g., website content) would not match the actual situation (e.g., a 404 error).
>
> Thank you again for your feedback. We hope our explanation has given you a better understanding of how TRACE handles task solvability.

---

> > ### Author Response · Authors · 2025-11-27
> >
> > Dear reviewer,
> >
> > Thank you for your insightful feedback, which is very helpful in improving the quality of our work. And we are also very grateful for the opportunity to further elaborate on our methodology
> >
> > We hope that the new analyses and expanded experiments added in the revision help clarify the concerns you raised. If there are any remaining questions or points that would benefit from further discussion, we would be more than happy to continue the conversation during the discussion phase. Your careful evaluation and encouragement have been very motivating for us, and we are sincerely grateful for your support. Please feel free to reach out to us if you have any questions or additional feedback. We welcome further discussion.
> >
> > Thank you once more!

---

### Author Response · Authors · 2025-12-01
**Full text of Reviewer H9KX’s updated (correct) review**

**Summary**: I apologize for any confusion in the earlier review. I have now carefully re-examined the paper and updated the comments, and based on the revised assessment, I think the original score does not need to be changed. This paper proposes a multi-agent framework for synthesizing more difficult and diverse questions from existing benchmarks. Besides, it also preserves the full exploration trajectories that led to their solutions, enabling trajectory validation and supporting reproducible benchmark evaluation. Experiments on GAIA show that the synthesized tasks' complexity are enhanced.

**Soundness**: 3: good

**Presentation**: 3: good

**Contribution**: 3: good

**Strengths**:

1. Saturation of existing benchmarks is a practical issue, addressing it is crucial for developing LLMs.
2. Previous methods for data synthesis generally lack support for reproducible benchmarks. This makes the proposed idea a novel contribution.
3. The experimental results show that the synthesized tasks indeed become substantially more challenging.

**Weaknesses**:
1. The authors does not analyze why model performance drops on the evolved tasks. Does the decline truly reflects increased difficulty? Or simply stems from the synthesized tasks drifting away from realistic scenarios.
2. Regarding the solvability of the synthesized tasks, it would benefit from additional empirical evidence to support the claim.

**Questions**: The authors state that evolved tasks are solvable, but I was confused how this is guaranteed? For example, if the original task uses website A but the evolved task requires non-exist website B, how do you detect and filter out such unsolvable cases?

**Flag For Ethics Review**: No ethics review needed.

**Rating**: 6: marginally above the acceptance threshold. But would not mind if paper is rejected

**Confidence**: 3: You are fairly confident in your assessment. It is possible that you did not understand some parts of the submission or that you are unfamiliar with some pieces of related work. Math/other details were not carefully checked.

**Code Of Conduct**: Yes

---

> ### Author Response · Authors · 2025-12-01
> **About the Special Issue of Reviewer H9KX's Review**
>
> Dear Area Chair,
>
> We would like to report a special circumstance. Reviewer H9KX mistakenly submitted a review intended for a different paper initially. Fortunately, the reviewer noticed and corrected the error promptly (modifying the review on Nov 20, 2025), so it had essentially no impact during the discussion phase. However, since a recent system rollback has re-displayed the original erroneous review, we feel it is necessary to explain this situation.
>
> To ensure the full integrity and fairness of the review process, we provides the correct version of the review. However, we recognize that **receiving the "correct" review directly from the authors could be a cause for concern**.
>
> Therefore, we kindly and earnestly request that you verify this sequence of events by checking the official **Revisions** of the review in the system. The logs should clearly show that the review was indeed updated by Reviewer H9KX. We believe this independent verification from the official record is crucial for a transparent and fair assessment of our paper.
>
> We have attached the correct review for your convenience, which should match the version you find in the system's history.
>
> Thank you for your understanding and attention to this critical matter.

---

### Author Response · Authors · 2025-12-01
**Rebuttal Summary and Explanation of Special Issues for the Area Chair**

Dear Area Chair,

We sincerely thank you for the time and effort you dedicated to the review, and we offer our profound respect for your commitment to academic fairness under these special circumstances. This document serves as our summary for the rebuttal phase, encompassing our responses to the reviewers' comments and further updates, which we hope facilitates your quick understanding.

**Special note**: Due to a system rollback, the currently displayed review from Reviewer H9KX is the initial, erroneous version. The reviewer had corrected this error promptly, which can be seen in **Revisions**. To avoid any misunderstanding, we are clarifying this situation here and will attach the correct review in another comment box.

**Initial reviews and acknowledged strengths**

Our paper initially received a mixed set of scores: **8 (Accept)** from Reviewer **s49g** (**confidence 5**), **6 (Bordline Accept)** from Reviewer **H9KX** (**confidence 3**), **4 (Borderline Reject)** from Reviewer **aYop** (**confidence 4**), and **2 (Reject)** from Reviewer **dJpA** (**confidence 4**).
While the scores varied, we are encouraged that **all four reviewers recognized the core insight, quality and value of our framework**. The positive reviewers (s49g, H9KX) praised our paper for its "**originality and relevance**," noting that the "**methodological soundness**" and modular design offer "**potential for community impact**". Reviewer H9KX highlighted that addressing benchmark saturation is "**crucial for developing LLMs**" and providing " **a novel contribution**". Reviewer aYop (initial score 4) noted the framework **provides an "automated mechanism" where the "validation stage ensures that all evolved tasks remain solvable"**. Reviewer dJpA (score 2), despite concerns on contribution, **rated Soundness and Presentation as "3: Good,"** explicitly stating the framework is "**carefully engineered and empirically validated**".

**Summary of rebuttal efforts and manuscript revisions**

We try to address all reviewer concerns through substantial revisions, including a major new benchmark experiment and deep statistical analyses. Our major efforts are summarized as follows:
1. **New experiments on Generalizability** (addressing concerns from Reviewer dJpA, s49g) To address the concern that experiments were "mostly centered on GAIA," we conducted a comprehensive new experiment applying TRACE to AIME-2024 (Competition Math).
 We demonstrated that TRACE successfully evolved math problems, resulting in a significant performance drop (e.g., Qwen3-235B accuracy dropped by 22.33%) and increased reasoning complexity (avg. token count increased by +8000).
 This proves TRACE is a domain-agnostic paradigm capable of evolving tasks from general agents to pure reasoning, directly resolving the scope concern.

2. **Deeper analysis of Difficulty and Construct Validity** (addressing concerns from Reviewer aYop, s49g, H9KX)
To confirm that performance drops reflect genuine difficulty rather than noise, we added rigorous statistical analyses:
- Token Length Analysis: We showed a consistent increase in reasoning tokens (e.g., +5065 tokens for Kimi-k2 on Round 2), confirming that evolved tasks force deeper search and planning.
- Capability Distribution: We quantified the shift in task demands, showing that TRACE triples the proportion of "Integrative Capabilities" (from 4.0% to 12.0%), validating that the benchmark evolves toward more realistic, complex problem-solving.
3. **Clarifications on Novelty and Efficiency** (addressing concerns from Reviewer dJpA)
We clarified the distinction between TRACE and prior works like AlphaEvolve and Self-Challenge Agents, highlighting our unique "trajectory-based exploration" approach. We also provided cost metrics (evolving AIME cost <$25 USD), dispelling concerns about computational overhead.

**Positive feedback and current status**

- **Reviewer aYop (Score raised 4 → 6)**: Following our new statistical analyses, this reviewer stated "You’ve largely resolved my concerns" and appreciated the clarifications on capability distribution, raising the score to 6. Additionally, we discuss the changes in the capability-requirement distribution with the reviewer.
- **Reviewer s49g (Score 8)**: Maintained a strong accept rating, encouraging us to "Keep up the good work!" and noting the community needs this type of contribution.
- **Reviewer H9KX (Score 6)**: Proactively corrected a submission error to confirm a positive rating.
- **Reviewer dJpA (Score 2)**: While this reviewer has not replied, **we have objectively addressed their primary critique regarding generality (via the new AIME experiment) and novelty (via the comparative analysis). Given their acknowledgement of the work's "Good" soundness and presentation, we believe the remaining divergence is manageable.**

Thank you again for your time and effort. We hope you find this summary and explanation helpful.

Best,

Submission8420 Authors

---

### Meta-Review · Area_Chair_ejRh · 2026-01-04

**Summary:**

The manuscript proposes to use a MAS to evolve the benchmark for the agent system. Idea-wise, it is quite interesting and crucial.

Though the ratings of the reviews differ (6, 4, 2, 8), after checking the reviews and rebuttal, the AC would recommend the acceptance.

**Reviewer Concerns:**

Reviewer H9KX questions (1) the model performance drop issue on the evolved tasks, and (2) the solvability of the synthesized tasks. The authors tried to address the concerns from the perspective of (i) the definition and metric of 'difficulty', (2) supplementary metric and experiments demonstrating increased difficulty, (iii) Qualitative Analysis of Realism for the Synthesis Task. The answer to the question of "the solvability of the synthesized tasks" is also satisfactory.

Reviewer aYop indeed had some similar concerns, regarding (1) whether the benchmark scores reflect meaningful aspects of model capability; (2) the missing detailed statistical analysis of the evolved tasks; (3) quantitative analyses of model capabilities after benchmark evolution; (4) how to interpret changes in the ranking order of models across the original and evolved benchmarks. After the rebuttal, Reviewer aYop agreed to raise the score to 6:
* The authors used the number of tokens generated by the models (e.g., when progressing from the original tasks to TRACE Evolved Rounds 1 and 2) to argue that the evolved tasks force models to engage in much longer chains of reasoning and tool use. The authors then examined how the capability distribution of tasks shifts before and after evolution.
* To answer the question on the authenticity and validity of evolved questions, the authors explained it from the perspective of the problem-formation and validation stages, with a case study.

Reviewer dJpA initially rated the manuscript with a score of 2, concerning the (1) distinct contribution relative to existing “agentic” methods, (2) limited exploration across other benchmarks or domains, and (3) the computational overhead introduced by the multi-agent pipeline (Proposer–Executor–Validator).
* The authors discussed the "Benchmark Self-Evolving" and "AlphaEvolve" within the landscape of agentic benchmark evolution. From the view of AC, the comparison with the Benchmark Self-Evolving is clear and can distinguish TRACE from the related work. However, the comparison with AlphaEvolve looks a bit strange.
* The authors additionally added results on AIME-2024, with a case study to compare the difference between the original task and the evolved task.
* The authors tried to address the computational overhead concern from the perspective of the Empirical Cost Report.

Reviewer s49g questions the (1) Selection bias — selecting on what you are reporting, (2) Construct validity — realism and meaningfulness of new tasks, and (3) Missing comparisons and broader context.
* The authors did not explicitly address the concern of selection bias, while the answers to the construction validity looks ok to the AC.

**Reviewer Scores:**

See context above.

---

### Decision · Program_Chairs · 2026-01-26

Accept (Poster)